



# Experimental development of a lake spray source function and its model implementation for Great Lakes surface emissions

Charbel Harb [1] and Hosein Foroutan [1]

[1]Department of Civil and Environmental Engineering, Virginia Tech, Blacksburg, VA, USA

**Correspondence:** Hosein Foroutan (hosein@vt.edu)

**Abstract.** Lake spray aerosols (LSAs) are generated from freshwater breaking waves in a similar mechanism to their saltwater counterparts, sea spray aerosols (SSAs). Unlike the well-established research field pertaining to SSAs, studying LSAs is an emerging research topic due to their potential impacts on regional cloud processes and their association with the aerosolization of freshwater pathogens. A better understanding of these climatic and public health impacts requires the inclusion of LSA
emission in atmospheric models, yet a major hurdle to this inclusion is the lack of a lake spray source function (LSSF), namely, an LSA emission parameterization. Here, we develop an LSSF based on measurements of foam area and the corresponding LSA emission flux in a marine aerosol reference tank (MART). A sea spray source function (SSSF) is also developed for comparison. The developed LSSF and SSSF are then implemented in the Community Multiscale Air Quality (CMAQ) model to simulate particle emissions from the Great Lakes surface from 10 to 30 November 2016. Measurements in the MART revealed that
the average SSA total number concentration was eight times higher than that of LSA. Consequently, the developed LSSF was around one and two orders of magnitude lower than the SSSF in the fine (r<0.2 $\mu$m) and coarse (r∼1-2 $\mu$m) aerosol size ranges, respectively. Model results revealed that LSA emission flux from the Great Lakes surface can reach ∼$10^5$ m$^{-2}$s$^{-1}$ during an episodic event of high wind speeds. These emissions only increased the average total aerosol number concentrations in the region by up to 1.65%, yet, their impact on coarse-mode aerosols was much more significant with up to a 19-fold increase in
some areas. The increase in aerosol loading was mostly near the source region, yet LSA particles were transported up to 1000 km inland. Above the lakes, LSA particles reached the cloud layer, where the total and coarse-mode particle concentrations increased by up to 3% and 98%, respectively. Overall, this study helps quantify LSA emission and its impact on regional aerosol loading and the cloud layer.

## 1 Introduction

In a similar mechanism to sea spray aerosols (SSAs) generation in saltwater (Lewis et al., 2004), lake spray aerosols (LSAs) can be produced by the entrainment of air bubbles by freshwater breaking waves and the subsequent bubble bursting process on the water surface (May et al., 2016). LSAs were first detected above the surface of the Laurentian Great Lakes in North America during an aircraft sampling campaign in summer 2009 (Slade et al., 2010), and have since become an emerging research topic (Axson et al., 2016; Chung et al., 2011; May et al., 2016, 2018a; Olson et al., 2019). Unlike SSAs which constitute a major
fraction of the global aerosol mass input into the atmosphere ($10^{12}$-$10^{14}$ kg y$^{-1}$; Textor et al. (2006)) and play a key role in





Earth's climate by affecting cloud properties and scattering light (Lewis et al., 2004), the role of LSAs in atmospheric processes is not well understood. While oceans cover around 70% of Earth's surface, freshwater lakes cover a significantly smaller area and are for the most part limited in fetch. Therefore, the impact of LSAs on atmospheric processes might be constrained to regional scales. Nonetheless, recent research has shown that LSAs are a vessel for the water-to-air dispersal of freshwater
bacteria (Harb et al., 2021), including cyanobacteria from harmful algal blooms (HABs) (May et al., 2018b; Olson et al., 2020; Plaas and Paerl, 2021), and hence might pose a risk to respiratory health. Moreover, LSAs have been sampled in the cloud layer above the Great Lakes surface (Olson et al., 2019), which indicates possible implications on cloud process and hence the regional climate.

Although breaking waves in saltwater and freshwater might look identical at first glance, looking more closely at the bubble
formation and bursting mechanisms reveals important differences between the two environments. At the subsurface level, the entrained bubble plume in saltwater is characterized by a higher void fraction and is comprised of smaller and more numerous bubbles than that in freshwater (Anguelova and Huq, 2018; Blenkinsopp and Chaplin, 2011; Harb and Foroutan, 2019; Scott, 1975). These differences have been ascribed to enhanced bubble coalescence in freshwater due to lower ionic content, which constrains the formation of the tiny bubble clouds observed in saltwater (Christenson et al., 2008; Hofmeier
et al., 1995). Disparities in bubble formation between freshwater and saltwater are manifested at the surface level, whereby saltwater whitecaps (foams) have been observed to be bigger and more persistent than their freshwater counterparts for the same wave breaking conditions (Harb and Foroutan, 2019; Monahan and Zietlow, 1969). Furthermore, saltwater whitecaps are comprised of a profusion of tiny surface bubbles, whereby those in freshwater contain bigger bubbles (Harb and Foroutan, 2019). Surface bubble size influences the spray aerosol ejection pathway, which can occur either during the shattering of
the bubble cap or after the ensuing cavity collapse. The former mechanism, known as film drop formation, occurs mostly in bubbles with a radius greater than 0.5-1 mm, while the latter, known as jet drop formation, occurs mostly in bubbles with a radius smaller than 0.5 mm (Deike, 2022; Lewis et al., 2004; Veron, 2015). Therefore, the smaller surface bubbles observed in saltwater whitecaps might enhance jet drop production in saltwater as compared to freshwater (Harb and Foroutan, 2019). Theses distinct air entrainment characteristics have important implications on the abundance and size of ejected SSAs and
LSAs. Laboratory experiments revealed that the ejection abundance of SSA is more than three times higher than that of LSA (Harb et al., 2021; May et al., 2016), and that the size distribution of freshly emitted SSAs is unimodal with an accumulation mode at 110 nm whereby that of LSAs is bimodal with an ultrafine mode at 46 nm and an accumulation mode at 180 nm (May et al., 2016). The aforementioned mechanistic differences in SSA and LSA production imply that they should be represented independently in general circulation models (GCMs) and chemical transport models (CTMs).

Due to their important role in Earth's climate, the inclusion of SSAs in GCMs and CTMs is an active research area (Barthel et al., 2019; Textor et al., 2006). Several SSA emission parameterizations, hereinafter sea spray source functions (SSSFs), have been proposed using both laboratory experiments and field measurements (de Leeuw et al., 2011; Lewis et al., 2004). These SSSFs essentially compute the number of SSA particles released per unit ocean area per unit time (O'Dowd and de Leeuw, 2007). The major driver of SSA emissions is wind stress, therefore, most SSSFs are formulated as a function of wind speed typ-
ically at reference height of 10 m ($u_{10}$) which is a common meteorological parameter in models (de Leeuw et al., 2011; Lewis





et al., 2004). However, it has been found that SSSF that rely solely on wind speed fail to predict measured SSA concentrations (Grythe et al., 2014; Jaeglé et al., 2011). Therefore, some SSSFs have been expanded to also include oceanic parameters such as sea surface temperature (SST) (Jaeglé et al., 2011; Mårtensson et al., 2003; Salter et al., 2015; Sofiev et al., 2011), water salinity (Sofiev et al., 2011), and wave state (Ovadnevaite et al., 2014), which led to better reproduction of observed SSA

concentrations.

A lake spray source function (LSSF), on the other hand, is still lacking to date which hampers our understanding of the atmospheric burden of LSAs. Chung et al. (2011) conducted the first ever modeling study of LSA emission from the surface of the Great Lakes using the mesoscale Weather Research and Forecasting model with online Chemistry (WRF-Chem). They reported up to a 20% increase in regional aerosol numbers above the lakes surface during July 2004 when LSA emissions

were enabled. However, it should be noted that they adopted an SSSF (Geever et al., 2005) to represent LSA emissions which is a significant source of uncertainty in that study (Chung et al., 2011). To improve on this simulation and understand the effect of LSAs on thermodynamic equilibrium in the Great Lakes region, Amiri-Farahani et al. (2021) conducted WRF-Chem simulations and found that calcium rich LSA particles lead to a 37% increase in particulate nitrate and a 16% decrease in particulate ammonium above the Great Lake surface. They used a corrected version of the Geever et al. (2005) SSSF by

scaling to the laboratory measurements of May et al. (2016). However, only correcting for the number emission flux when adapting an SSSF for LSA emissions might not be sufficient, since the same wind speed over freshwater and saltwater does not induce the same wave breaking conditions. With these two studies being the only LSA modeling studies to date, it is clear that more modeling work is needed to better understand the effect of LSAs on atmospheric processes, specifically in the Great Lakes region. Such studies would be much improved if an LSSF was made available to the community, rather than having to

use corrected versions of SSSFs.

Here, we develop the first LSSF starting from laboratory experiments using the widely adopted marine aerosol reference tank (MART; Stokes et al. (2013)), which is now considered the de facto experimental method for generating realistic spray aerosols (Mayer et al., 2020). We also use the MART system to develop an SSSF for comparison. To test the developed LSSF, we use the Community Multiscale Air Quality (CMAQ) model to simulate LSA emissions from the surface of the Great Lakes.

With a combined surface area of 244,000 km$^2$, these lakes form collectively the largest inland body of unfrozen freshwater on Earth (Gronewold et al., 2013). The Great Lakes basin is home to 48.5 million people (2011 figures; Méthot et al. (2015)) and is considered to be a critical component of the economic health of central North America since it supports a wide array of commercial, industrial, and recreational activities (Wuebbles et al., 2019). Therefore, the Great Lakes were chosen for these simulations due to their sheer size (sometimes referred to as "inland seas"; Sterner et al. (2017)), their proximity to major

population centers in central North America, and their susceptibility to high wind speeds and wave breaking (Axson et al., 2016; Monahan and Zietlow, 1969; Slade et al., 2010).





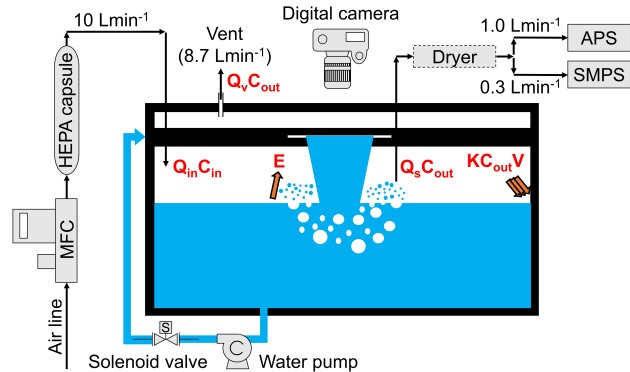

**Figure 1.** A schematic of the experimental setup used in this study.

## 2 Methods

### 2.1 Water samples collection and preparation

A solution of synthetic freshwater, based on Lake Michigan ionic concentrations (Chapra et al., 2012; May et al., 2016), was produced by dissolving anhydrous inorganic salts (Fisher Scientific; $CaCO_3 \geq 99\%$, $MgSO_4 \geq 99\%$, NaCl $\geq 99\%$, KCl $\geq 99\%$) in ultrapure water ($\sim 18.2$ M$\Omega$.cm, Picopure®) to achieve the following concentrations: 1 mM $Ca^{2+}$, 1 mM $CO_3^{2-}$, 0.4 mM $Mg^{2+}$, 0.4 mM $SO_4^{2-}$, 0.3 mM $Na^+$, 0.3 mM $Cl^-$, and 0.02 mM $K^+$. This water sample was used to develop the LSSF.

Synthetic seawater was prepared by dissolving artificial sea salt (Instant Ocean® Spectrum Brands, Blacksburg, VA, USA) in ultrapure water ($\sim 18.2$ M$\Omega$.cm, Picopure®) with a 35 g/kg mixing ratio corresponding to a typical marine salinity. This water sample was used to develop the SSSF.

To investigate LSA production from natural freshwater with organic contents, two freshwater samples (180 L each) were also collected from the surface of Claytor Lake (Pulaski County, VA, USA) using pre-autoclaved 20 L HDPE carboys (Fig. S1). To contrast seasonality and biological activity, the first sample was collected in the fall on 31 October 2020 whereas the second sample was collected in the summer on 9 August 2021. During sampling, water temperature and salinity were measured using an Extech EC170 (Extech Instruments, Nashua, NH, USA) salinity-temperature meter and are reported in Table S1. The collected water samples were then immediately transported to the laboratory to be used within 24 hours after collection.

### 2.2 Experimental development of the source functions

#### 2.2.1 Aerosol generation and size distribution measurements

Spray aerosols were generated using a custom-built MART (Stokes et al. (2013); see Fig. 1). In short, the setup is comprised of a polycarbonate tank (100 cm x 54.6 cm x 61 cm) with two concentric tubes at the top of the tank that allow water to exit as a uniform sheet (Fig. 1). A 1/3 HP self-priming utility pump (AMT Pumps, Royersford, PA, USA) allows the water to circulate in the system. Water sheet intermittency can be controlled by a Parker skinner valve (Parker Hannifin, Madison,



MS, USA) mounted on the pump discharge port and connected to a Macromatic TR-53122-07 time delay relay (Macromatic, Menomonee Falls, WI, USA). More details about the setup construction and operation can be found in Harb and Foroutan

(2019). Similar to the MART (Stokes et al., 2013; Prather et al., 2013), the setup used in this study has been shown to reproduce the correct physical characteristics of air entrainment and spray aerosol generation found in oceanic breaking waves, and has been previously used to generate laboratory SSAs and LSAs (Harb and Foroutan, 2019; Harb et al., 2021).

To generate spray aerosols from the water samples described in Sect. 2.1, a total volume of 147 L from each water sample was added to the MART providing a water depth of 27 cm. The water sheet was operated continuously for 4 h at a flow rate

of 24.5 $\mathrm{Lmin}^{-1}$ to maximize aerosol production. It is important to note that continuous air entrainment might lead to biases in the size of ejected spray aerosols if surface foam evolution is suppressed (Harb and Foroutan, 2019), yet the size of the tank was large enough to minimize the interaction between surface bubble rafts and tank walls (see Fig. S2 in the supplement). Prior to aerosol size distribution measurements, the headspace was flushed with HEPA-filtered air until the background particle concentration was less than 10 $\mathrm{cm}^{-3}$. Nascent spray aerosols ejected from the generated foam patch were sampled ~2 cm

above the water surface, and then directed to a scanning mobility particle sizer (SMPS 3936, TSI, Shoreview, MN, USA) and an aerodynamic particle sizer (APS 3321, TSI, Shoreview, MN, USA) (Fig. 1) to determine particle size distribution. It is a common practice to include a dryer upstream of the aerosol sizing instrumentation to measure dry particle size (Fuentes et al., 2010; May et al., 2016; Stokes et al., 2013), however, there is concern about supermicron particle losses in such setups (Salter et al., 2014). To examine these losses, two aerosol size distribution measurements were taken in the experiments with

synthetic freshwater and saltwater solutions. In the first measurement, aerosols were directly sampled by the aerosol sizing instrumentation (SMPS and APS) without drying. In the second measurement, aerosols were dried using a diffusion dryer (TSI 3062, TSI, Shoreview, MN, USA) installed directly upstream of the aerosol sizing instrumentation (Fig. 1). Experiments with the Claytor Lake water samples were carried out without a dryer. Therefore, six sets of experiments were carried out in total. More details about these experiments can be found in Table S2.

The SMPS was operated at a sampling flow rate of 0.3 $\mathrm{Lmin}^{-1}$ and a scan rate of 5 min, providing a size distribution of particles with mobility diameter ($d_m$) between 14 and 700 nm. The APS was operated at a sampling flow rate of 1.0 $\mathrm{Lmin}^{-1}$ and a scan rate of 5 min, providing a size distribution of particles with aerodynamic diameter ($d_a$) between 0.5 and 20 $\mu$m. The inlet flow rate of clean air was maintained at 10 $\mathrm{Lmin}^{-1}$ using an Aalborg GFCS-010013 mass flow controller (Aalborg Instruments & Controls, Orangeburg, NY, USA) and a vent in the tank lid allowed excess air flow (8.7 $\mathrm{Lmin}^{-1}$) to escape

(Fig. 1). In order to obtain a single aerosol size distribution spanning the SMPS and APS measurement ranges, the $d_m$ and $d_a$ size distributions were merged into a single physical diameter ($d_p$) size distribution using a procedure described elsewhere (Khlystov et al., 2004; Stokes et al., 2013; May et al., 2016). Mobility diameters measured by the SMPS were converted to physical diameters by assuming spherical particle geometry:

$$d_p = d_m \tag{1}$$

Aerodynamic diameters measured by the APS were converted to physical diameters using the following relation:





$$d_p = \frac{d_a}{\sqrt{\frac{\rho_{eff}}{\rho_0}}} \tag{2}$$

In Eq. (2), $\rho_0$ is equal to unit density (1 gcm$^{-3}$) and $\rho_{eff}$ is an effective density assigned to particles sized by the APS. For both LSAs and SSAs, $\rho_{eff}$ was considered to be equal to 1.5 g/cm$^3$ (May et al., 2018a; Moffet et al., 2008) assuming considerable aerosol liquid content (relative humidity RH mostly greater than 90%; see Table S2). When stitching, particle bins in the overlapping size range of the SMPS and APS were removed due to uncertainties in particle counting efficiency (Stokes et al., 2013).

### 2.2.2 Foam area determination

To monitor the evolution of the foam patch area generated inside the MART, a Nikon D750 camera was used to take photographs of the water surface during active air entrainment in the synthetic freshwater and saltwater solutions. Due to condensate accumulation on the inside of the tank walls, it was not possible to capture foam photographs concurrently with aerosol size distribution measurements. Therefore, these photographs were taken in subsequent air entrainment experiments. The same water flow conditions described in Sect. 2.2.1 were used, and each experiment lasted for approximately 2 h with the camera (Fig. 1) programmed to capture a single photograph every 10 min.

To determine the foam patch area, photographs were analyzed using the image processing software ImageJ (Schneider et al., 2012). Foam areas were identified manually and were sized after scaling the photographs using pictures of a precision ruler placed on the water surface. An example of a processed surface foam image is shown in Fig. S2.

### 2.2.3 Source functions derivation

The continuous whitecap method (CWM; Monahan and Callaghan (2015)) was used to determine source function formulations from the experiments with synthetic freshwater (hereinafter simply LSSF) and synthetic saltwater (hereinafter simply SSSF) samples. In brief, the CWM infers the production flux of spray aerosols from measurements of size-depended spray aerosol production and scaling it per whitecap area. An inherent assumption in this method is that a whitecap area has the same production rate of spray aerosols regardless of its generation method (e.g., in situ breaking wave or laboratory water sheet) (de Leeuw et al., 2011; Monahan and Callaghan, 2015). Using this approach, the source function formulation reads:

$$\frac{\partial F}{\partial r}(u_{10}, r) = W(u_{10}).\frac{\partial F_{wc}}{\partial r}(r) \tag{3}$$

In Eq. (3):

– $\partial F/\partial r$ ($m^{-2}s^{-1}\mu m^{-1}$) is the rate of spray aerosol generation, per unit area of water surface, per unit increment of spray droplet radius $r$.



- $W$ $(m^{-2}m^{-2})$ is the whitecap coverage defined as the area of whitecap foam per unit area of water surface. $W$ is usually parameterized as a function of $u_{10}$ (Anguelova and Webster, 2006). For our formulations, we adapted the commonly used Monahan and Muircheartaigh (1980) parameterization for saltwater whitecap coverage. To account for reduced foaming in freshwater as compared to saltwater, and in the absence of a freshwater whitecap parameterization to date, a factor $\alpha$ was introduced to the saltwater parameterization of Monahan and Muircheartaigh (1980). Following the proposition of Monahan (1971), $\alpha$ was defined as the ratio of foam exponential decay time constant ($\tau$) in freshwater to that in saltwater, and was calculated using previously published $\tau$ values in freshwater and saltwater measured using the MART (Harb and Foroutan, 2019, Table 1). Interestingly, the calculated $\alpha$ value of 0.65 from these MART experiments is in excellent agreement with the 0.66 value calculated by Monahan (1971) from their whitecap simulation tank experiments (Monahan and Zietlow, 1969). Therefore, the corrected whitecap coverage W reads ($\alpha = 1$ for saltwater, and $\alpha = 0.65$ for freshwater) :

$$W(u_{10}) = \alpha(3.84 \times 10^{-6}u_{10}^{3.41}) \tag{4}$$

- $\partial F_{wc}/\partial r$ $(m^{-2}s^{-1}\mu m^{-1})$ is the number of aerosol particles produced per unit of whitecap area per unit time as a function of spray droplet radius $r$. $\partial F_{wc}/\partial r$ was determined experimentally by dividing the measured steady state, size-resolved, number emission rate of spray aerosols inside the MART headspace $E$ by the foam (whitecap) area on the water surface $A$, as follows:

$$\frac{\partial F_{wc}}{\partial r}(\mu m^{-1}s^{-1}m^{-2}) = \frac{E(\mu m^{-1}s^{-1})}{A(m^2)} \tag{5}$$

In Eq. (5), $A$ was determined from the foam imaging experiments described in Sect. 2.2.2, whereas $E$ was determined by considering a mass-balance (Eq. (6) and Fig. 1) inside the MART headspace under the assumption of well-mixed conditions (Quadros and Marr, 2011; Lin and Marr, 2017).

$$\frac{d(C_{out}V)}{dt} = Q_{in}C_{in} - Q_vC_{out} - Q_sC_{out} + E - kVC_{out} \tag{6}$$

Under steady state conditions, defined as the period with less than 20% variation in total aerosol number concentration in the MART headspace, the left-hand-side of Eq. (6) becomes zero, and $E$ can be calculated from Eq. (7).

$$E = -Q_{in}C_{in} + Q_vC_{out} + Q_sC_{out} + kVC_{out} \tag{7}$$

In Eqs. (6) and (7), $Q_{in}$ is the inlet flowrate of HEPA-filtered air, $C_{in}$ is the concentration of spray aerosols in the inflow (equal to zero), $Q_v$ is the flow rate of excess air that is vented, $C_{out}$ is the measured size-resolved number concentration of spray aerosols in the headspace, $Q_s$ is the sampling flowrate of the aerosol sizing instrumentation (SMPS+APS), $V$ is the headspace volume, and $k$ is the wall loss coefficient (Fig. 1). $k$ was determined experimentally by arresting water flow and spray aerosol generation in the tank (i.e., $E= 0$), and then measuring the decay of $C_{out}$ with time. Wall losses were assumed to be a first-order exponential decay process and $k$ was calculated by fitting an exponential function to the measured $C_{out}$ decayed over time. More details about the wall loss coefficient determination can be found in Sect. S1 and Fig. S3 in the supplement.





A common convention is to report a source function formulation in terms of the particle radius at a reference relative humidity of 80% (de Leeuw et al., 2011). Therefore, the measured particle radius was converted to the value it would have at an RH=80% (i.e., $r_{80}$) using the correction proposed by Zhang et al. (2006, Eq. (2)). The experimentally determined source functions, now expressed as $\partial F/\partial r_{80}$, were then fitted and formulated as the sum of two lognormally distributed modes, as follows:

$$\frac{\partial F}{\partial r_{80}}(u_{10}, r_{80}) = \alpha(3.84 \times 10^{-6} u_{10}^{3.41}) \sum_{i=1}^{2} \frac{N_i}{\sqrt{2\pi}\ln(\sigma_i)} \exp(-\frac{1}{2}\frac{(\ln r_{80} - \ln \mu_i)^2}{(\ln \sigma_i)^2}) \tag{8}$$

    In Eq. (8), $u_{10}$ is expressed in meters per second, $r_{80}$ is expressed in micrometers, and $N_i$, $\sigma_i$, and $\mu_i$ are the number production flux, the geometric standard deviation, and the geometric mean of the i-th mode, respectively.

## 2.3   Model implementation

    To test the developed LSSF, LSA emissions from the surface of the Great Lakes system in North America were considered.

The Community Multiscale Air Quality (CMAQ) model version 5.3 (CMAQv5.3; Appel et al. (2021)) was used for this purpose. The simulation time period (10 to 30 November 2016 with a 9 day spin-up) was chosen to coincide with the season of minimal to no lake ice cover (Wang et al., 2012) and high wind speeds over the surface of the lakes (Li et al., 2010). Simulations were performed using CMAQv5.3 benchmark test case of the Conterminous United States (CONUS) (US EPA, 2019). In brief, this test case employs a 12-km uniform horizontal grid covering the CONUS, parts of northern Mexico and

southern Canada, and the eastern Pacific and western Atlantic oceans, with 35 vertical layers expanding up to 50 hPa. Meteorological inputs are provided by a WRFv3.8 simulation and were processed using the Meteorology-Chemistry Interface Processor (MCIPv5.0; Appel et al. (2021); Otte and Pleim (2010)). The physics parameterizations used in the WRFv3.8 simulation include the Morrison double-moment microphysics scheme (Morrison et al., 2009), the Rapid Radiative Transfer Model for General circulation models (RRTMG) radiation scheme (Iacono et al., 2008), the Kain–Fritsch convective parameteri-

zation (Kain, 2004), the Pleim-Xiu land-surface model (Pleim and Xiu, 2003; Xiu and Pleim, 2001), and the Asymmetric Convective Mixing 2 planetary boundary layer model (Pleim, 2007). Baseline anthropogenic emissions were provided by the 2016beta Emission Modeling Platform inventory (EMP; http://views.cira.colostate.edu/wiki/wiki/10197). Boundary conditions were provided from a hemispheric CMAQ (HCMAQ) simulation with a 108 x 108 km polar stereographic grid covering the northern hemisphere, 44 vertical layers, and meteorological fields from WRFv3.8. Science configurations used in CMAQv5.3

include the updated M3dry model for deposition, the CB06r3 chemical mechanism and AERO7 aerosol model for atmospheric chemistry, and the KMT version 2 (KMT2) and the KMTBR modules for cloud chemistry (Appel et al., 2021). More details about CMAQv5.3 settings and evaluation can be found in Appel et al. (2021).

    The current spray aerosols emission scheme in CMAQv5.3 only allows for SSA emission from the surface of saltwater bodies (i.e., eastern Pacific and western Atlantic oceans in the present case). The SSA scheme uses the Gong (2003) source

function with $\theta = 8$ for online SSA emission flux calculations with a linear SST dependence following the Ovadnevaite et al. (2014) parameterization (Gantt et al., 2015). For the purpose of this study, the Gong (2003) source function was replaced

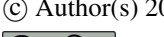



**Table 1.** Summary of the three emission scenarios: BASE, LAKE, and SEA.

| Simulation | Ocean emissions | Great Lakes emissions |
|------------|-----------------|-----------------------|
| BASE | SSSF | None |
| LAKE | SSSF | LSSF |
| SEA | SSSF | SSSF |

by our synthetic saltwater source function, with the SST dependence kept the same. Concurrently, spray aerosol emissions from the Great Lakes surface were enabled and were evaluated for three emission scenarios. In the BASE scenario, no LSA emissions from the Great Lakes surface were considered (default CMAQv5.3 configuration for the CONUS). In the LAKE and

SEA simulations, LSA emissions from the Great Lakes surface were enabled using the developed LSSF and SSSF, respectively, with no lake surface temperature (LST) dependence. In both of the latter scenarios, spray aerosols emitted from the Great Lakes surface were modeled as chemically-inert dry particles with a density of 1.5 gcm$^{-3}$ (May et al., 2018a; Moffet et al., 2008). The emission scenarios evaluated in this study were designed to assess the contribution of LSA emissions to regional aerosol loading in the Great Lakes basin (LAKE scenario), and the overestimation of LSA emissions brought about by considering the

Great Lakes as saltwater bodies and using an SSSF (SEA scenario) (see e.g., Chung et al. (2011)). A brief summary of these emission scenarios is shown in Table 1.

## 3   Results and Discussions

### 3.1   Spray aerosol size distribution

The average size distributions of spray aerosols generated in the MART headspace during the last 2 h of active air entrainmnet

in each experimental set are shown in Fig. 2a. It is obvious from this figure that the abundance of spray aerosols generation in saltwater is significantly higher than that in freshwater, with an average SSA and LSA total number concentrations (wet) of 822 and 102 cm$^{-3}$, respectively. The higher generation of spray aerosols in saltwater compared to that in freshwater concurs with previous observations (Harb et al., 2021; May et al., 2016), and can be attributed to higher void fractions and whitecap formation following wave breaking in saltier waters (Anguelova and Huq, 2018; Harb and Foroutan, 2019; Scott, 1975). The

shape of the SSA and LSA size distributions are also distinct. In saltwater, the size distribution of wet SSAs exhibits two distinct modes at 0.09 and 2.3 $\mu$m, whereas that of dry SSAs exhibits a single mode at ~0.2 $\mu$m with the second supermicron mode being suppressed (Fig. 2a). While the dry SSA size distribution agrees well with previous laboratory measurements in the MART (Prather et al., 2013; Stokes et al., 2013), the second distinct supermicron mode observed for wet SSAs has not been previously reported. We speculate that this second mode is a "jet drop" mode (Harb et al., 2021; Mårtensson et al., 2003),

and is not detected when using a dryer due to tubing losses which will be discussed later on. In freshwater, on the other hand, all LSA size distributions are characterized by a single dominant mode at ~0.1 $\mu$m regardless of their source (i.e., synthetic

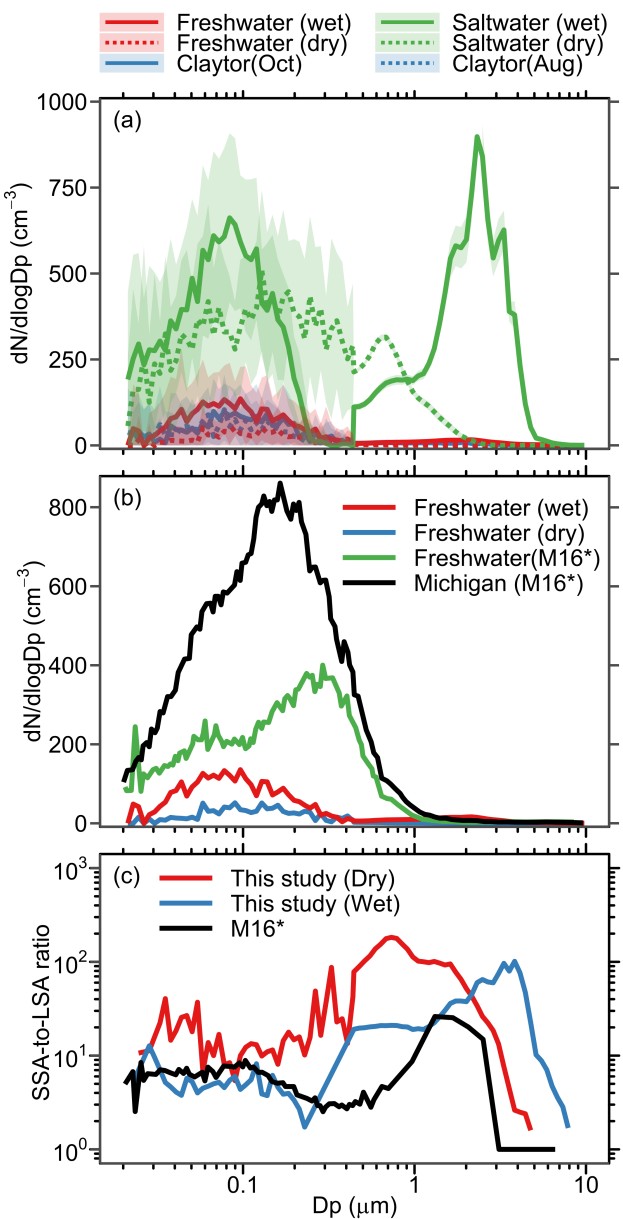

**Figure 2. (a)** Average aerosol size distribution generated in the MART using the synthetic freshwater, synthetic saltwater, and Claytor Lake water samples collected in October and August. Shaded areas represent ± 1 standard deviation. "dry" and "wet" denote measurements made with and without a dryer, respectively. **(b)** LSA size distribution from synthetic freshwater (wet and dry) plotted along with LSA size distributions from synthetic and Lake Michigan freshwater from May et al. (2016). **(c)** Ratio of SSA-to-LSA aerosol size distributions produced in the MART (wet and dry) and that produced in the LSA generator of May et al. (2016).

\* M16 denotes May et al. (2016)





or Claytor Lake water). The subtle variation in LSA size distributions between synthetic and natural lake water could be due to low biological activity in the water samples collected from Claytor Lake in October and August. However, high biological content in lake water has been observed to significantly increase LSA production abundance (Olson et al., 2020). Therefore,

lakes with high seasonal variability in biological content, in particular those with algal bloom occurrences, might exhibit large temporal variations in LSA emissions abundance.

To the best of our knowledge, only one laboratory study (May et al., 2016) attempted to investigate differences between LSA and SSA production to date. Figure 2b shows the LSA size distribution in synthetic freshwater (wet and dry) plotted along with LSA size distributions from synthetic and Lake Michigan freshwater produced by an LSA generator (May et al., 2016). The

LSA generator, a small water tank ($\sim$18 L) with four circular water jets, is inherently different from the MART system used in this study. Therefore, it is not valid to comment on differences in the magnitude of number concentrations between the two studies and, hence, the comparison is limited to the shape of the size distributions. Comparing the LSA size distribution from synthetic freshwater in both studies reveals a unimodal distribution in the MART at $\sim$0.15 and $\sim$0.09 $\mu$m for dry and wet LSAs, respectively. In the LSA generator, on the other hand, the LSA size distribution is bimodal with a minor mode at

0.08 $\mu$m and a major mode at 0.3 $\mu$m. It is likely that these disparities are mostly due to different LSA generation methods as the synthetic freshwater solution is identical in both studies. The size distribution of LSAs generated from Lake Michigan freshwater in the LSA generator, in contrast, is mostly unimodal at 0.18 $\mu$m, which is close to the major mode (at $\sim$0.15 $\mu$m) observed in wet LSAs produced from synthetic freshwater in the MART.

To better comment on the relative magnitude of LSA and SSA production in the two studies, Fig. 2c compares the SSA-

to-LSA number size distribution ratio (hereinafter referred to as SSA-to-LSA ratio) measured from synthetic freshwater and saltwater solutions in the MART (wet and dry) to that measured in the LSA generator (dry) of May et al. (2016). The SSA-to-LSA ratio for wet aerosols in the MART and that for dry aerosols in the LSA generator show good agreement up to 0.1 $\mu$m, with values ranging from 4 to 9. Meanwhile, the SSA-to-LSA ratio for dry aerosols in the MART exhibited higher values of up to 37 in this size range. In the 0.1-10 $\mu$m particle size range, noticeable disparities are observed between the SSA-to-LSA

ratios. In the accumulation mode (0.1-0.5 $\mu$m), the SSA-to-LSA ratio for wet aerosols in the MART is not reliable due to uncertainties in the measurement efficiency of the SMPS in this size range (see Fig. 2a), therefore, we limit this discussion to supermicron particles (i.e., $D_p$>1 $\mu$m). While the SSA-to-LSA ratios for dry aerosols in MART and dry aerosols in the LSA generator drop significantly after for $D_p$>1 $\mu$m, this ratio for wet aerosols in the MART exhibits a peak of 900 at $\sim$2.3 $\mu$m, which is driven by the distinct supermicron mode in the wet SSA size distribution shown in Fig. 2a. It is worth noting that the

SSA-to-LSA ratio in the LSA generator has been employed by Amiri-Farahani et al. (2021) to determine an LSSF by scaling the Geever et al. (2005) SSSF, which underscores its importance in comparing the magnitude of LSA and SSA production fluxes at different aerosol particle sizes.

The effect of including a dryer upstream of the aerosol sizing instrumentation on the aerosol size distributions, especially on the SSA size distribution, is evident in Fig. 2a. To further analyze this effect, Fig. 3 shows the synthetic saltwater and freshwater

average aerosol number (a,b), surface area (c,d), and volume (e,f) size distributions of wet and dry aerosols, corresponding to sampling with and without a dryer, respectively. As mentioned previously, the dryer at our disposal was a TSI 3062 diffusion

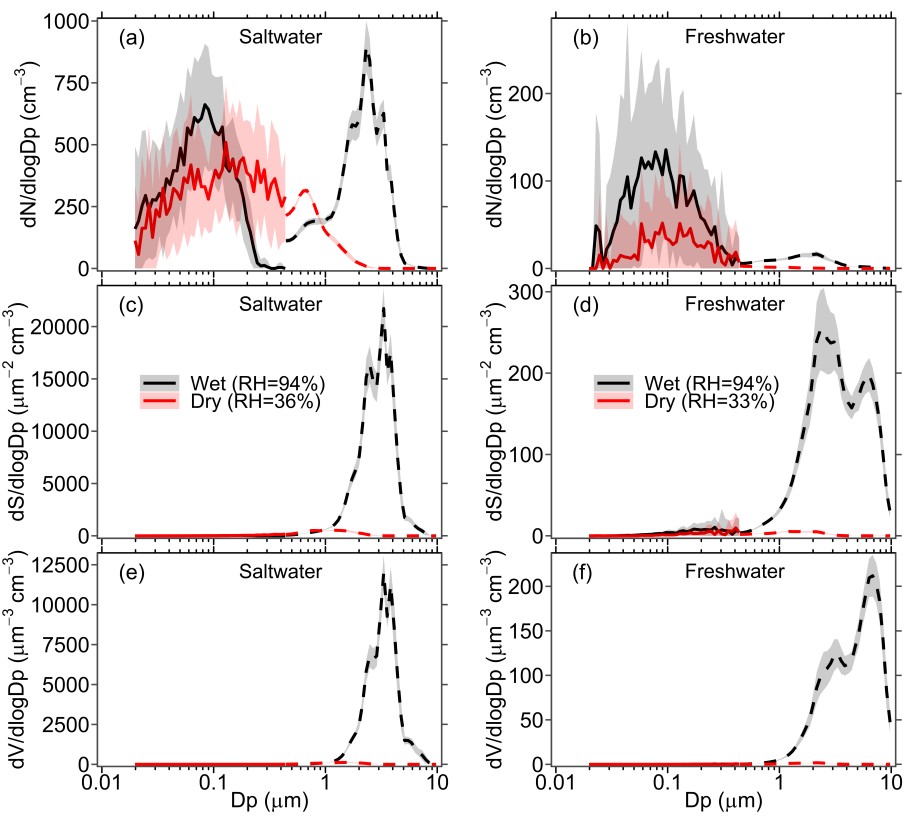

**Figure 3.** The synthetic saltwater and freshwater average aerosol **(a,b)** number , **(c,d)** surface area , and **(e,f)** volume size distributions of dry and wet aerosols, corresponding to sampling with and without a dryer, respectively. Shaded areas represent ± 1 standard deviation.

dryer, with two Swagelok® 90°-elbows at the inlet and outlet ports. As it turned out, drying the particles before sampling was not trivial, and led to considerable tubing losses, particularly in the supermicron size range. In saltwater (Fig. 3a), the supermicron mode is completely lost when drying the particles and some losses are also observed for submicron particles where the peak shifts from 0.09 to 0.2 $\mu$m. The loss in submicron particles is more evident in freshwater where the number concentration peak of 136 (±93) cm$^{-3}$ at 0.1 $\mu$m for wet aerosols is reduced by more than half to 51 (±92) cm$^{-3}$ at 0.13 $\mu$m when drying the particles. The issue of particle loss when including a dryer in the sampling line was raised previously by Salter et al. (2014), who estimated a 50 % loss for particles with dry diameter greater than 5 $\mu$m using the von der Weiden et al. (2009) procedure. Using this same approach, we attempted to estimate particle losses in a greatly simplified tubing configuration of the TSI 3062 diffusion dryer (see Sect.S2 and Fig. S4a in the supplement). We find that in the submicron size range, particle loss was less than 10%, yet this loss increases exponentially for supermicron particles reaching more than 50% for particles with a diameter greater than 5 $\mu$m (Fig. S4b). Hence, it is likely that particle losses in the dryer were even more considerable, especially in the supermicron size range, which explains the loss of the supermicron peak in the SSA size distribution when drying the particles (Fig. 3a). Yet, in the absence of a dryer, there is a discontinuity in the wet aerosol size distribution in the





**Table 2.** Lognormal parameters for the present LSSF and SSSF. Refer to Eq. (8).

|  | LSSF | SSSF |
| --- | --- | --- |
| $\alpha$ | 0.65 | 1 |
| $N_1$ | $6.4106 \times 10^7$ | $1.0452 \times 10^8$ |
| $N_2$ | $1.2140 \times 10^5$ | $2.3646 \times 10^6$ |
| $\mu_1$ | 0.0137 | 0.0167 |
| $\mu_2$ | 0.5852 | 0.6815 |
| $\sigma_1$ | 2.4623 | 2.6022 |
| $\sigma_2$ | 1.5694 | 1.4096 |

overlapping size range between the SMPS and APS (Fig. 3a). Moreover, there is a sharp decrease in the wet aerosol number concentrations in the upper size range of the SMPS (i.e., $D_p \sim$0.13-0.40 $\mu$m). This sharp decrease might be associated with our observation of water accumulation in the impactor inlet in the absence of a dryer, which might reduce the impactor cut-off size. Therefore, caution is required when interpreting results in this size range. Losses in surface area and volume concentrations are even more severe, since supermicron particles are especially relevant for these quantities. Indeed, the peak in surface area

concentration for $D_p$>1 $\mu$m drops from 21722 ($\pm$ 1964) to 520 ($\pm$ 66) $\mu$m$^{-2}$cm$^{-3}$ in saltwater and from 253 ($\pm$ 52) to 5.4 ($\pm$ 0.8) $\mu$m$^{-2}$cm$^{-3}$ in freshwater. Similarly, the peak in volume size distribution drops from 12025 ($\pm$1087) to 120 ($\pm$26) $\mu$m$^{-3}$cm$^{-3}$ in saltwater, and from 212 ($\pm$23) to 1.8 ($\pm$0.34) $\mu$m$^{-3}$cm$^{-3}$ in freshwater. Given these considerable losses, source function development in the following section (Sect. 3.2) will be based on wet aerosol measurements from synthetic freshwater and saltwater solutions.

**3.2   Source function development**

Using the procedure described in Sect. 2.2.3, we developed an LSSF and an SSSF from wet aerosol measurements in the MART using the synthetic saltwater and freshwater solutions, respectively. The lognormal parameters for each formulation (see Eq. (8)) are given in Table 2. These source functions are plotted in Fig. 4a for $u_{10} = 10$ ms$^{-1}$. Due to the aforementioned (Sect. 3.1) uncertainty in the SMPS counting efficiency of wet aerosols in the accumulation mode size range (corresponding

to r$_{80} \sim$ 0.1-0.2 $\mu$m), data points in this size range were excluded from the SSSF fit. Figure 4a reveals that the SSA emission number flux is one order of magnitude higher than that of LSA for $r_{80}$<0.2 $\mu$m, and almost two orders of magnitude higher for $r_{80}$=0.2-2 $\mu$m. Unlike the coarse mode which is similar between the LSSF and the SSSF (0.72 vs 0.77 $\mu$m), the fine mode of the SSSF centered at 0.042 $\mu$m is greater than that of the LSSF, which is centered at 0.031 $\mu$m. This fine mode in the LSSF compares well with the Aitken mode (0.025-0.035 $\mu$m) measured above the Great Lakes surface (Slade et al., 2010).

The developed LSSF and SSSF are compared to a collection of common SSSFs from literature in Fig. 4b. It is worth noting that some source functions shown in this figure are reported as a function of dry particle diameter (D$_{dry}$) (e.g., Clarke et al.,





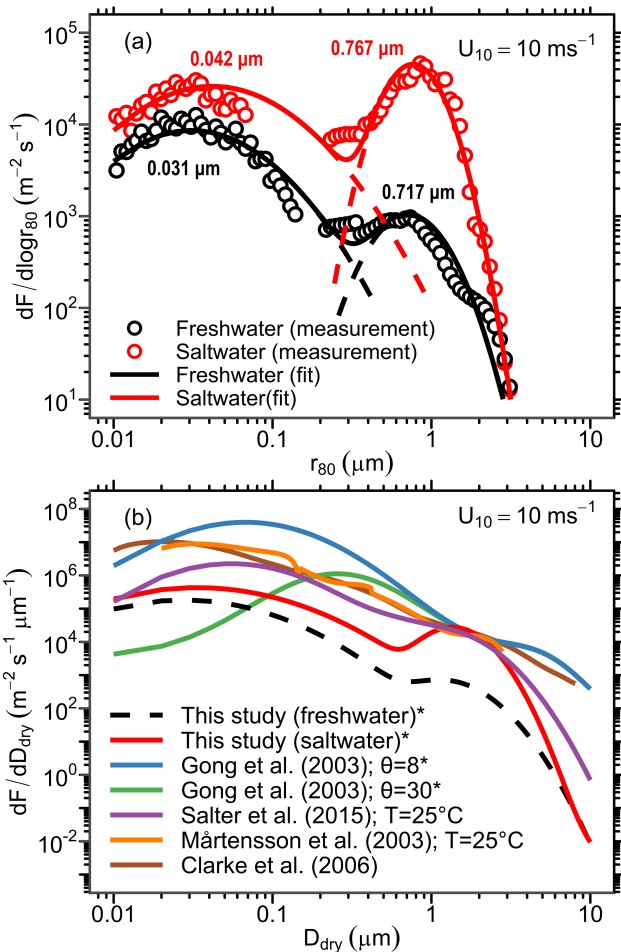

**Figure 4. (a)** The developed LSSF (freshwater) and SSSF (saltwater) plotted for $u_{10} = 10\ \mathrm{ms}^{-1}$. Data points represent the measured emission parameterizations using the MART setup, solid curves represent the lognormal distribution fit, and dashed lines represent each lognormal mode. **(b)** Comparison between the present LSSF and SSSF and a collection of common SSSFs from literature for $u_{10} = 10\ \mathrm{ms}^{-1}$. Note the change in axes between panels **(a)** and **(b)**.

\* Corrected from their original formulation as a function of $r_{80}$ by assuming $r_{dry} = r_{80}/2$

2006; Mårtensson et al., 2003; Salter et al., 2015), while others (e.g., Gong, 2003) are reported in terms of particle radius at RH=80% ($r_{80}$). For the sake of consistency, we, therefore, corrected the latter parameterizations (denoted by an asterisk in the legend) to become a function of dry particle diameter $D_{dry}$ by assuming $r_{dry} = r_{80}/2$, a common rule of thumb (O'Dowd and de Leeuw, 2007; Veron, 2015). We start by comparing the SSSFs to assess the validity of our method for developing sound estimates of SSA emission fluxes. It is evident from Fig. 4b that there is a general agreement between all SSSFs in the supermicron size range. In the submicron size range, however, there is some disagreement between the SSSFs which span up to 3 orders of magnitude in the ultrafine range ($D_{dry}$<0.1 $\mu$m). Yet, all SSSFs exhibit a distinct accumulation mode at around






0.1 to 0.2 $\mu$m. The discrepancy in the SSSFs magnitudes in the submicron size range can be attributed to different methods

for developing the emission parameterization in each study. The Gong (2003) parameterization, for instance, is a mathematical extension of the Monahan et al. (1986) parameterization to diameters below 0.2 $\mu$m. This extension, nevertheless, is just an adjustable mathematical formulation (using a parameter $\theta$) for setting the shape of the source function for the sub-0.2 $\mu$m size range, and lacks therefore a scientific rationale for its development (O'Dowd and de Leeuw, 2007). The Clarke et al. (2006) source function is developed based on ambient measurements of SSAs generated from the surfzone, and hence, might

overestimate SSA emission from open ocean breaking waves. Meanwhile, the Mårtensson et al. (2003), the Salter et al. (2015), and the present SSSF are developed using measurements of laboratory generated SSAs. Nevertheless, the method in which SSAs were generated in each study is different, with the Mårtensson et al. (2003) study employing a small chamber (2 L) with a glass frit, the Salter et al. (2015) study using a larger cylindrical tank ( 170 L) with a circular water jet, and the present study using the $\sim$300 L MART with a thin water sheet. Moreover, the current SSSF and the Mårtensson et al. (2003) source functions

use the Monahan and Muircheartaigh (1980) formulation for whitecap coverage dependence on wind speed, whereas the Salter et al. (2015) source function employs a formulation of the air entrainment flux dependence on wind speed modified from Long et al. (2011).

To further assess the validity of the here derived source functions, we estimate the emission mass flux as a function of wind speed using the different source function formulations shown in Fig. 4b. We compare these estimates to field measurements

of submicron SSA emission mass flux (PM$_1$), obtained using an aerosol mass spectrometer (AMS), which estimates the mass of particles with a vacuum aerodynamic diameter D$_{va}$=0.5-1 $\mu$m, or D$_{dry}$=r$_{80}$=0.029–0.580 (Ceburnis et al., 2008, 2016; Ovadnevaite et al., 2012). Assuming spherical particles, we use Eq. (9) and Eq. (10) to estimate the mass flux from source functions expressed in terms of particle radius at RH=80% ($\overline{F}_{80}$) and from source functions expressed in terms of dry particle diameter ($\overline{F}_{dry}$), respectively:

$$\overline{F}_{80} = \frac{4}{3}\pi\rho_{80} \int\limits_{r_{80,1}}^{r_{80,2}} \frac{dF}{r_{80}} r_{80}^3 dr_{80} \tag{9}$$

$$\overline{F}_{dry} = \frac{1}{6}\pi\rho_{dry} \int\limits_{D_{dry,1}}^{D_{dry,2}} \frac{dF}{dD_{dry}} D_{dry}^3 dD_{dry} \tag{10}$$

Where, $\rho_{80}$ and $\rho_{dry}$ denote the wet (RH=80%) and dry particle density, and are assumed to be equal to 1.5 and 2.16 g/cm$^3$, respectively. The limits of integration were chosen to be $r_{80,1}=D_{dry,1}=0.029$ $\mu$m and $r_{80,2}=D_{dry,2}=0.580$ $\mu$m to match the measurement range of the AMS instrument. Figure 5 shows the calculated mass fluxes compared against the PM$_1$ measure-

ments of Ceburnis et al. (2008, 2016) on a (a) linear, and (b) logarithmic y-axis. As expected, the emission mass flux computed from the present LSSF is, at least, one order of magnitude lower than that computed from the SSSFs at any wind speed (Fig. 5b). Furthermore, this comparison revealed that the present SSSF and that of Salter et al. (2015) agree relatively well with the field measurements. Meanwhile, the Gong (2003), Mårtensson et al. (2003), and Clarke et al. (2006) SSSFs overestimate the measured PM$_1$ flux (Fig. 5b) as previously reported (Ceburnis et al., 2016; Ovadnevaite et al., 2012; Salter et al., 2015).





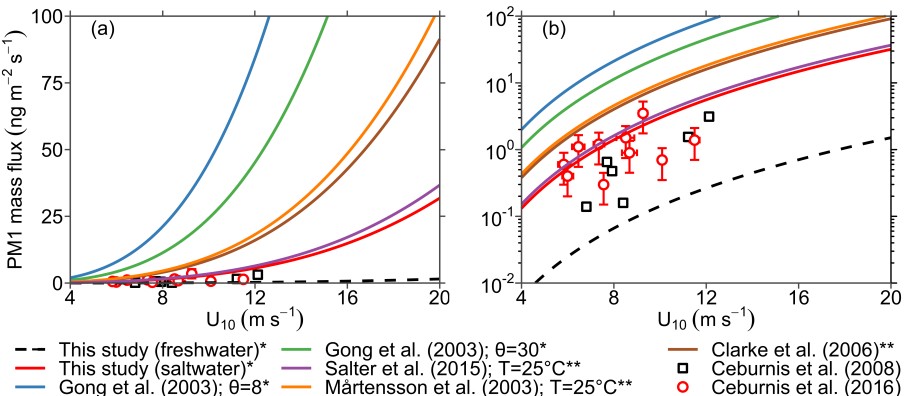

**Figure 5.** The calculated mass fluxes from the source functions shown in Fig. 4b compared with $PM_1$ mass flux measurements from Ceburnis et al. (2008, 2016) on a **(a)** linear, and **(b)** logarithmic y-axis.

* Computed using Eq. (9)

** Computed using Eq. (10)

As discussed in the de Leeuw et al. (2011) review, these SSSFs also appear to overpredict submicron number emission flux, as show in Fig. 4b, and hence fail to agree with SSA number concentrations measured in the marine boundary layer. This overprediction is likely to become more drastic at even higher wind speeds (>12 ms$^{-1}$), as suggested by Fig. 5a, which raises concern about the skill of many CTMs and GCMs that use such source functions for their SSA emission schemes (de Leeuw et al., 2011; Textor et al., 2006).

## 3.3 Model simulation

As described in section 2.3, we implemented the newly developed LSSF in the CMAQ model to assess LSA emission from the Great Lakes surface for the simulation scenarios shown in Table 1. We start this discussion by exploring the LSA emission abundance during significantly windy conditions over the Great Lakes surface. Figure 6 shows the modeled number emission flux of LSA particles during an episode of very high 10-m wind speeds (19 November 2016, 15:00:00 UTC) for the LAKE and

SEA scenarios. During this time, winds were generally northwesterly over Lakes Superior and Michigan and southwesterly over the remaining lakes, with wind speeds ranging from a high of 17 to 21 ms$^{-1}$ over most of Lakes Superior and Michigan, and a low of 5 to 9 ms$^{-1}$ over Lake Ontario (Fig. 6a). A clear dependence of LSA number emission flux on wind speed, as anticipated, can be seen in Fig. 6b, c. The highest emissions are from the surface of Lakes Superior and Michigan and range between $7\times10^4$ and $1\times10^5$ m$^{-2}$ s$^{-1}$ in the LAKE scenario, and between $3\times10^5$ and $5\times10^5$ m$^{-2}$ s$^{-1}$ in the SEA scenario.

Meanwhile, emissions from Lake Ontario, for instance, were up to two orders of magnitude lower, ranging from $1\times10^3$ to $7\times10^3$ m$^{-2}$ s$^{-1}$ in the LAKE scenario and $4\times10^3$ to $3\times10^4$ m$^{-2}$ s$^{-1}$ in the SEA scenario. The results of Fig. 6 reveal that the LSA emission flux is highly sensitive to wind conditions, increasing exponentially with higher wind speeds as shown in Fig.





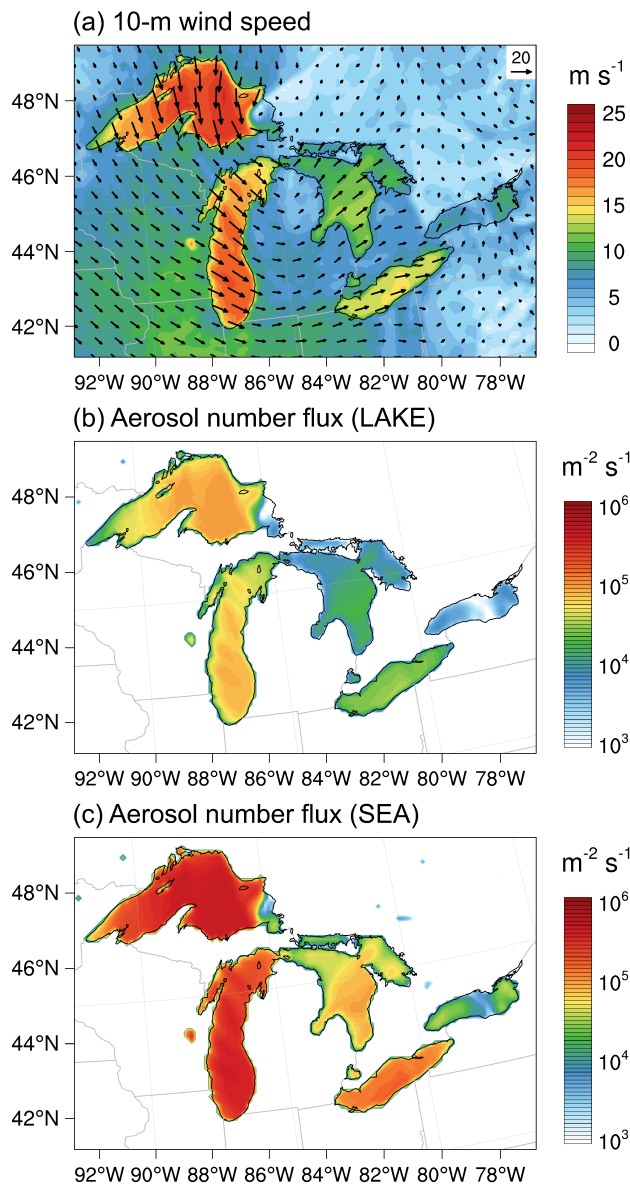

**Figure 6. (a)** The 10-m wind speed and the corresponding aerosol number emission flux from the Great Lakes surface in the **(b)** LAKE and **(c)** SEA scenarios on 19 November 2016, 15:00:00 UTC.

5a. Furthermore, using an SSSF to represent LSA emissions (i.e., the SEA scenario) can overapproximate the actual number emission flux by up to one order of magnitude.

While looking into episodic events of very high wind speeds highlights the extent of LSA emission from the Great Lakes surface, a more holistic understanding requires studying long-term averaged emissions. Figure 7 shows the time-averaged total



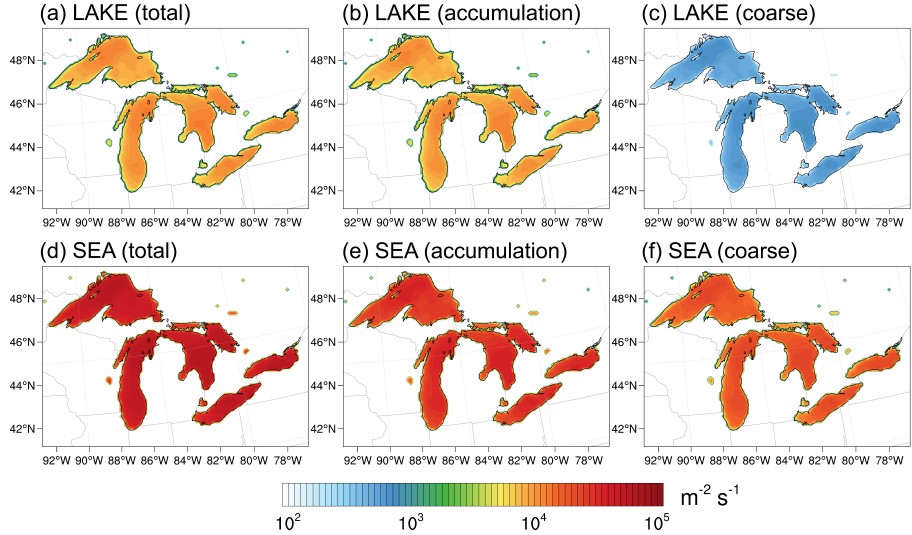

**Figure 7.** Average (10-30 November 2016) **(a,d)** total, **(b,e)** accumulation-mode, and **(c,f)** coarse-mode aerosol number emission flux in the LAKE and SEA scenarios.

(a,d), accumulation-mode (b,e), and coarse-mode (c,f) number emission fluxes of particles from the Great Lakes surface for the entire simulation period using the LAKE and SEA scenarios. In the LAKE scenario, the total number emission flux ranges from $7 \times 10^3$ to $1.3 \times 10^4$ $m^{-2}$ $s^{-1}$ (Fig. 7a). For comparison, the average emission rates in the simulation of Amiri-Farahani et al. (2021) for the month of November 2015 were on the order of $10^6$ $m^{-2}$ $s^{-1}$, which are two orders of magnitude higher than the results of this simulation despite their use of an improved LSSF formulation. Particle emission in the LAKE scenario is dominated by the accumulation mode (94% contribution), with coarse-mode particles only contributing to 6% ($3 \times 10^2$ to $7.5 \times 10^2$ $m^{-2}$ $s^{-1}$) of total emissions (Fig. 7b, c). In the SEA scenario, on the other hand, the average total aerosol number emission flux ranges from $3 \times 10^4$ to $6.5 \times 10^4$ $m^{-2}$ $s^{-1}$ (Fig. 7d), which leads to a significant 4-fold overapproximation of actual emissions. Accumulation-mode particles also contribute the most (62%) to this emission (Fig. 7e), yet, coarse-mode particles also contribute significantly (38%) (Fig. 7f) unlike their low contribution in the LAKE scenario. The contribution of each particle size mode to particle emissions in Fig. 7b, c, e, and f mirror their relative magnitudes in Fig. 4a, whereby coarse-mode particles are of comparable magnitude to the accumulation-mode particles in the SSSF, whereas they are one order of magnitude lower in the LSSF.

In the remainder of this section, we discuss the implication of spray aerosol emissions from the Great Lakes surface on regional aerosol number and mass concentrations. It is important to reiterate that these spray aerosols are essentially considered to be chemically-inert particles with a density of 1.5 $gcm^{-3}$ (see Sect.2.3). Such a consideration facilitates the tracking of these particles in the atmosphere without chemical processing. However, the chemistry involving LSA particles is important as it has been shown that these particles can alter thermodynamic equilibrium in the Great Lakes region, leading to an increase in particulate nitrate and a decrease in particulate ammonium (Amiri-Farahani et al., 2021). Therefore, without a realistic





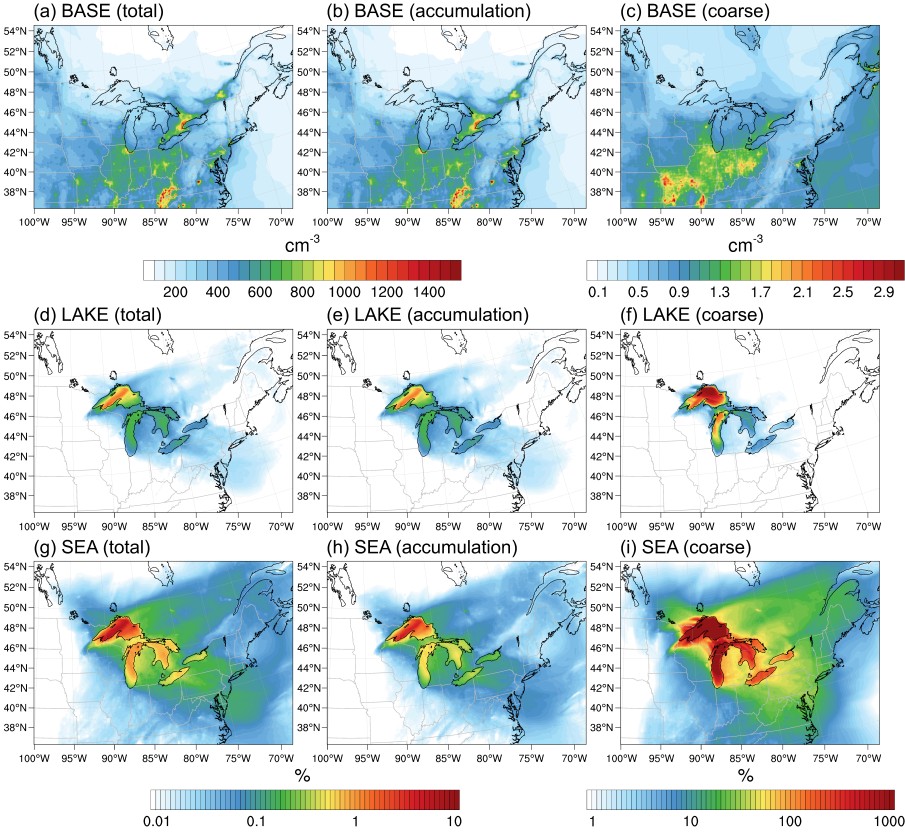

**Figure 8.** Average (10-30 November 2016) **(a-c)** total, accumulation-mode, and coarse-mode surface-layer aerosol number concatenations in the BASE scenario, and their corresponding percent increase in the **(d-f)** LAKE and **(g-i)** SEA scenarios.

chemical speciation, the results of this simulation only provide a preliminary estimate of the impact of particle emission from the Great Lakes surface on regional aerosol loading.

Figure 8 shows the total, accumulation-mode, and coarse-mode surface-layer aerosol number concatenations in the BASE scenario averaged over the simulation period (a-c) and their corresponding percent increase in the LAKE (d-f) and SEA scenarios (g-i). In the absence of surface emissions from the Great Lakes (i.e., BASE scenario), regional aerosol loading in the Great Lakes basin is dominated by anthropogenic emissions from the Chicago and Toronto metropolitan areas. In these regions, total number concentrations reach more than 1000 cm$^{-3}$, while average concentrations above the Great Lakes surface are mostly smaller than 500 cm$^{-3}$ (Fig. 8a). Looking at the contribution of each mode reveals a clear dominance of accumulation-mode particles on regional aerosol population, as expected. Meanwhile, aerosol number concentrations in the coarse mode are three orders of magnitude lower than those in the accumulation mode, reaching ~2 cm$^{-3}$ in the Chicago and Toronto metropolitan areas and <1 cm$^{-3}$ above the Great Lakes surface (Fig. 8b, c). When enabling LSA emissions from the Great Lakes surface (i.e., LAKE scenario), the increase in the average total (and accumulation-mode) aerosol number concentrations





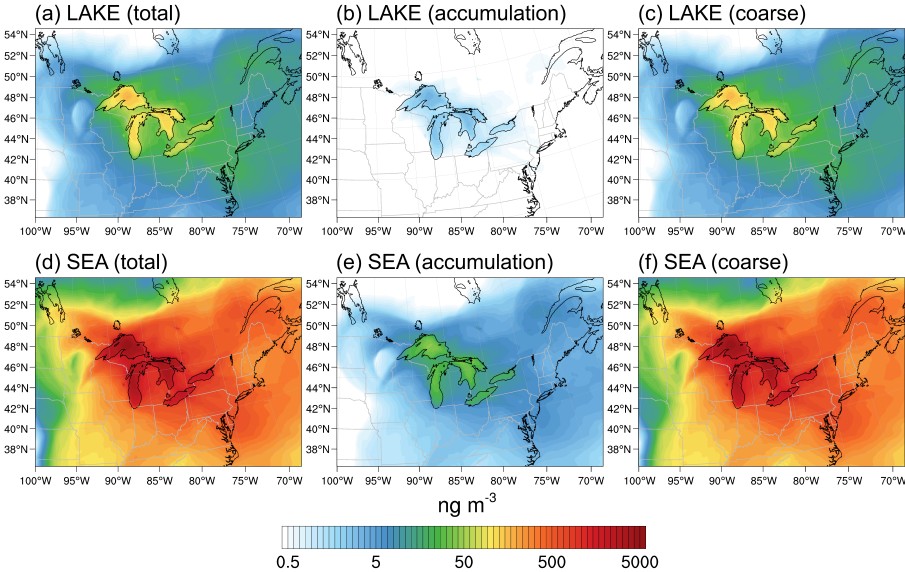

**Figure 9.** Average (10-30 November 2016) total, accumulation-mode, and coarse-mode surface-layer aerosol mass concatenations of particles emitted from the Great Lakes surface in the **(a-c)** LAKE and **(d-f)** SEA scenarios.

is mostly in the source region (i.e., above the lakes surface), with up to 1.65% in northwestern Lake Superior and <0.25% average increase above other parts of the lakes (Fig. 8d, e). A much more prominent increase can be seen for coarse-mode

particles, for which the percent increase can reach up to 1900% in northwestern Lake Superior and ranges from 5 to 150% over other parts of the lakes (Fig. 8f). This apparent increase can be attributed to low preexisting aerosol concentrations (<1 cm$^{-3}$) in the source region, especially over the remote northern lakes (Fig. 8c), coupled with discernible LSA emissions in the coarse mode from the lakes on the order of $10^2$ m$^{-2}$ s$^{-1}$ (Fig. 7c). Enabling SSA emissions from the Great Lakes surface (i.e., SEA scenario), on the other hand, leads to a more noticeable increase in regional aerosol loading. Average total number

concentrations increase by up to 7.5% in northwestern Lake Superior (Fig. 8g, h), which is lower than the maximum increase of 20% reported by Chung et al. (2011) over the same region. Over other parts of the lakes, the increase in average total aerosol number concentrations ranges from 0.5 to 1.2%. Furthermore, the percent increase in coarse-mode aerosol number concentrations is much more significant and ranges between 90% in western Lake Erie and up to 64000% in northwestern Lake Superior. Inland, coarse-mode aerosol number concentrations reach more than 10% in regions up to ~1000 km northeast

of the lakes (Fig. 8i). Altogether, using an SSSF led to around one order of magnitude overapproximation of LSA contribution to regional aerosol numbers.

Figure 9 shows the average total, accumulation-mode, and coarse-mode surface-layer aerosol mass concatenations of particles emitted from the surface of the Great Lakes in the LAKE (a-c) and SEA scenarios (d-f). In the LAKE scenario, the average mass concentration is highest in the source region (50 to 175 ngm$^{-3}$) and can reach more than 10 ngm$^{-3}$ inland up to ~1000

km east and northeast of the Lakes (Fig. 9a). This inland transport of LSA particles supports previous field observations of





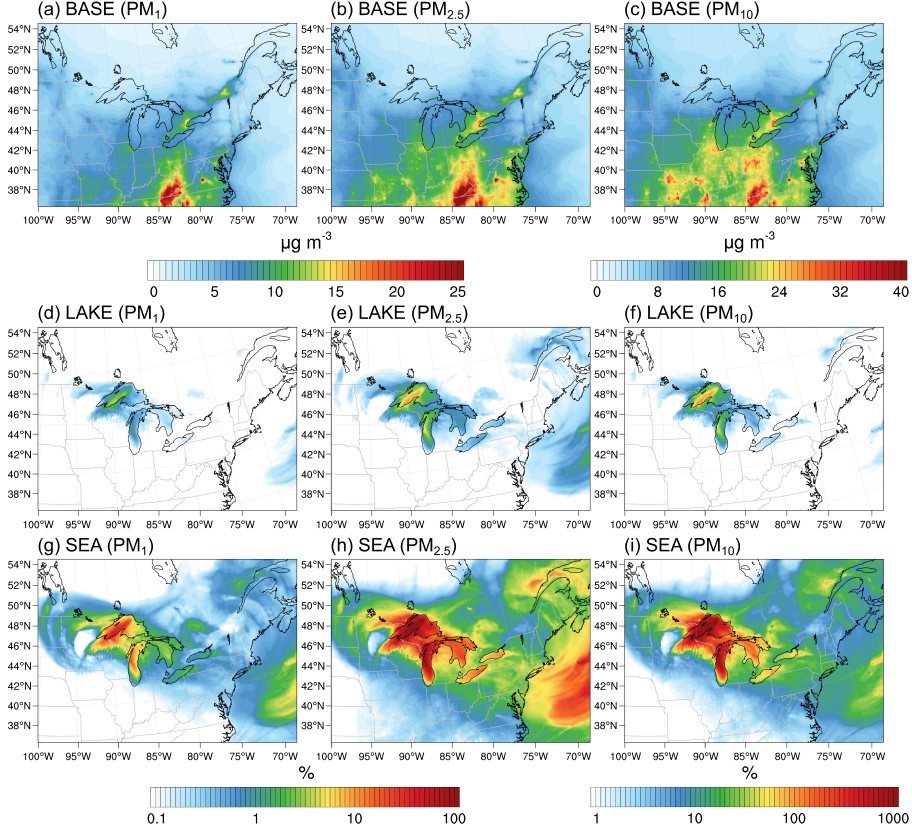

**Figure 10.** Average (10-30 November 2016) **(a-c)** PM$_1$, PM$_{2.5}$, and PM$_{10}$ surface-layer concatenations in the BASE scenario, and their corresponding percent increase in the **(d-f)** LAKE and **(g-i)** SEA scenarios.

LSA contribution to the aerosol population in a rural site in northern Michigan located >25 km from the nearest Great Lakes source (May et al., 2018a). Looking more closely at the contribution of each mode reveals that coarse-mode particles dominate the mass concentration ($\sim$ 98%) (Fig. 9c), as expected, whereby the contribution of accumulation-mode particles is only $\sim$2% (less than 4 ngm$^{-3}$) and is mostly constrained to the source region (Fig. 9b). In the SEA scenario, the average mass concentra-
tion reaches up to 5160 ngm$^{-3}$ in the source region and 300 ngm$^{-3}$ inland with significantly greater spatial coverage (Fig.9d). As with the LAKE scenario, the average mass concentration is also dominated ($\sim$ 99%) by coarse-mode particles (Fig. 9f), whereby the contribution of accumulation-mode particles is only $\sim$1% (less than 50 ngm$^{-3}$) (Fig. 9e). Overall, it can be seen from Fig. 9 that the mass concentration of particles emitted from the Great Lakes surface is overapproximated by more than one order of magnitude when using an SSSF instead of an LSSF.

To put these mass emissions into perspective, Fig. 10 shows regional PM$_1$, PM$_{2.5}$, and PM$_{10}$ surface-layer concentrations in the BASE scenario averaged over the simulation period (a-c) and their corresponding percent increase in the LAKE (d-f) and SEA scenarios (g-i). In the BASE scenario, PM$_1$, PM$_{2.5}$, and PM$_{10}$ concentrations above the Great Lakes surface were highest





over the southern parts of Lake Michigan and Lake Erie where they were about 7, 9, and 15 $\mu$gm$^{-3}$, respectively, and were lowest in the remote northern lakes, specifically over northwestern Lake Superior where they were about 3,4, and 6 $\mu$gm$^{-3}$,

respectively (Fig. 10a-c). Hotspots of PM concentrations can be clearly seen in the the Chicago and Toronto metropolitan areas where PM$_1$, PM$_{2.5}$, and PM$_{10}$ average concentrations reach 15, 20, and 28 $\mu$gm$^{-3}$, respectively. When LSA emissions from the surface of the Great Lakes are enabled (i.e., LAKE scenario), PM$_1$ and PM$_{2.5}$ increase by up to 3% and 10% in northwestern Lake Superior, respectively, driven by low preexisting PM concentrations in that area (Fig. 10d-e). The average percent increase in PM$_{10}$ is rather more significant reaching 117 % over northwestern Lake Superior (Fig. 10f). Overall, the increase in PM

concentrations in the LAKE scenario is mostly in the source region with some increase inland specifically in the vicinity of the lakes. On the other hand, SSA emissions (i.e., in the SEA scenario) result in up to 45%, 250%, and 3000% average increase in PM$_1$, PM$_{2.5}$, and PM$_{10}$ concentrations in the source region, respectively (Fig. 10g-i). Therefore, using an SSSF to represent LSA emissions resulted in one order of magnitude overapproximation of Great Lakes surface emission contribution to regional PM concentrations. Interestingly, it can also be seen from Fig. 10 that the effect of surface emissions from the

Great Lakes can extend far beyond the source region and into the western Atlantic Ocean. These increases in faraway regions stem from an episodic event of high particle emissions from the Great Lakes surface on 20-21 November 2016, followed by atmospheric transport to the western Atlantic Ocean which is otherwise an area with low preexisting PM concentrations (Fig. 10a-c). Therefore, emissions from the Great Lakes surface can extend further inland during episodic events of very high wind speeds, a feature that is somehow concealed when averaging over several weeks.

We conclude this section by examining the vertical reach of particles emitted from the Great lakes surface and their potential to reach the cloud layer. Figure 11 shows vertical cross-sections (slices) of the percent increase in total aerosol number concentration in the LAKE (a,c,e,g) and SEA (b,d,f,h) scenarios during an episodic event of high wind speed on 19 November 2016, 15:00:00 UTC. Also shown is the model estimated location of cloud bottom and top layers for reference. It is clear from this figure that emitted spray aerosols can reach several kilometers above the water surface, and into the cloud layer, which

concurs with previous field measurements showing LSA particles incorporation into Great Lakes clouds (Olson et al., 2019). In the LAKE scenario, the highest percent increase in vertical aerosol number concentration is ∼5% up to 1000 m above ground level and occurs in the middle of slice 2, which falls into the central region of the Great Lakes (Fig.11c). Moving further North or South from the midst of the lakes (i.e, slices 1, 3, and 4) reduces the influence of surface emissions on vertical aerosol number concentrations, with the percent increase in aerosol number concentrations above 100 m being mostly smaller than 1%

(Fig.11 a,e,g). In the cloud layer, this percent increase ranges from less than 1% and up to 3%. In the SEA scenario, on the other hand, the contribution of the Great Lakes to vertical aerosol number concentration becomes noticeably higher (Fig. 11 b,d,f,h), reaching 26% up to 1000 m above ground level in slice 2. In the cloud layer, the percent increase of aerosol number concentration ranges from less than 1% and up to 13%. Therefore, using an SSSF led to around 4-fold overapproximation of LSA contribution to vertical aerosol number concentrations.

Given the small contribution (<3%) of LSA particles to the aerosol number concentration in the cloud layer (Fig. 11), it might seem that LSA emissions from the Great Lakes surface have a limited influence on regional cloud processes. However, the size of the ejected spray aerosol plays a major role in its ability to act as seed particle for cloud drops, with recent studies





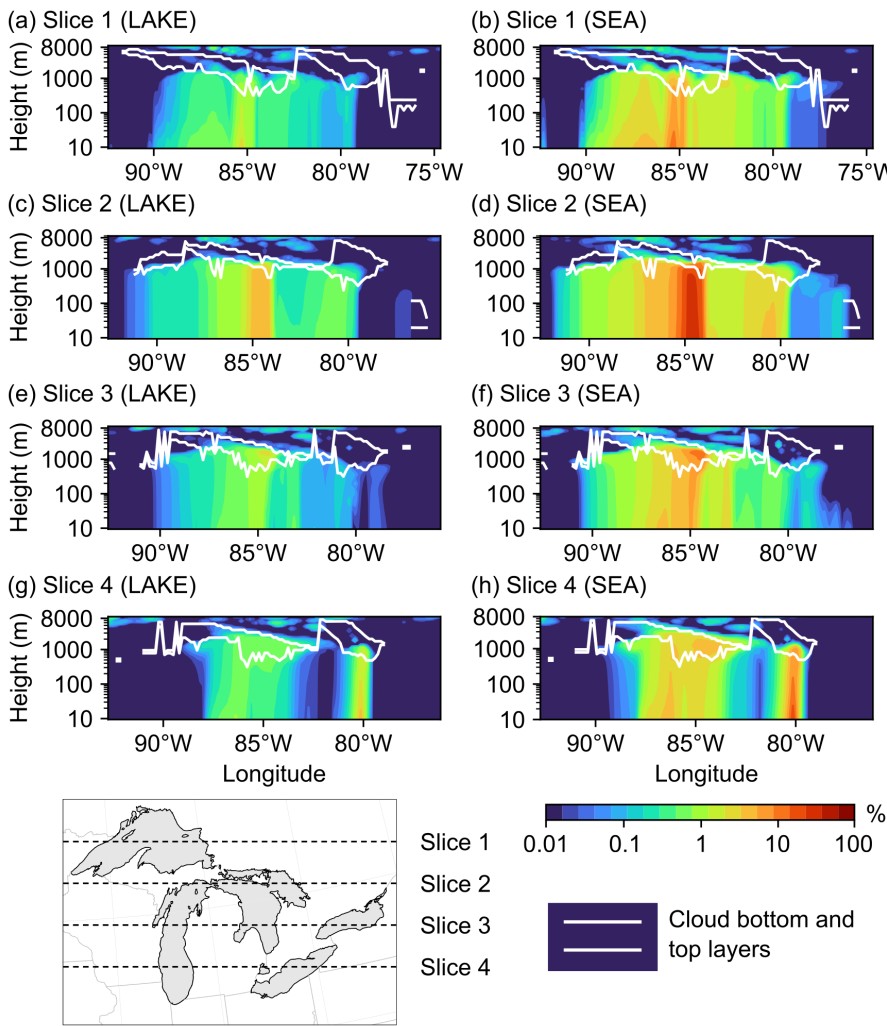

**Figure 11.** Percent increase in the total vertical aerosol number concentrations in 4 cross-sections (slices) spanning the Great Lakes basin from North to South in the **(a-c-e-g)** LAKE and **(b-d-f-h)** SEA scenarios on 19 November 2016, 15:00:00 UTC. Also shown are the model estimated cloud bottom and top layers in each slice.

indicating that the majority of ice nucleating particles (INPs) originating from SSAs are supermicron in scale (Creamean et al., 2019; Mitts et al., 2021). It is likely that these results translate to LSA particles, therefore, Fig. 12 shows the same data presented in Fig.11 but for coarse-mode particles. It is evident from this figure that the increase in coarse-mode particles in the LAKE scenario is significant. In slice 2, for instance, these particles increase by up to 144% up to 1000 m above ground level (Fig. 12c). When moving away from the midst of the Lakes (Fig. 12 a,e,g), the increase in coarse particles above 100 m is still significant and ranges from 5% to 100%. In the cloud layer, and across all slices, the increase in coarse-mode particle concentrations is significant and ranges from less than 1% all the way up to 98%. When considering the unrealistic scenario





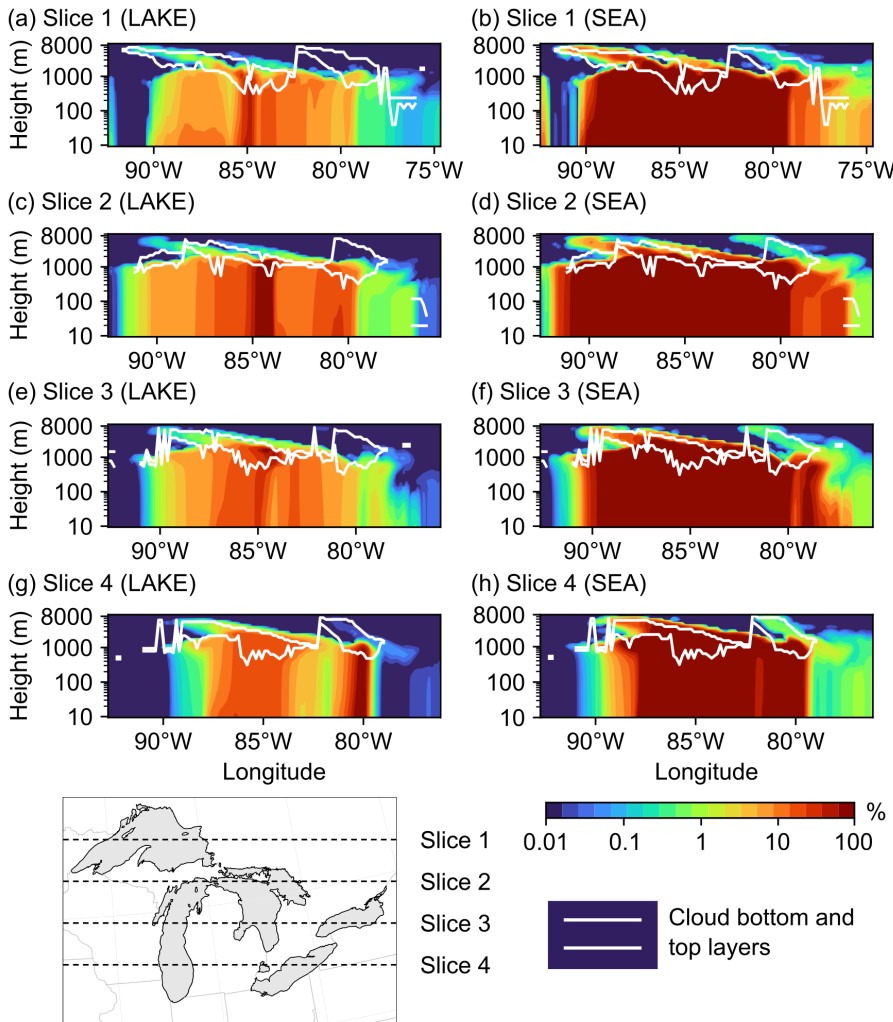

**Figure 12.** Same as Fig.11 but for coarse-mode particles

of SSA emissions from the Great Lakes surface (i.e., SEA scenario), there is a drastic percent increase in coarse-mode particle concentrations, with values exceeding 100 % across all slices (Fig. 12 b,d,f,h). Given that freshwater is considered an important INP reservoir (Moffett et al., 2018) and that clouds are highly sensitive to the presence of even low INPs concentrations (Rosenfeld et al., 2014), the significant increase of up to 98% in coarse-mode particles in the cloud layer indicates that LSA emissions from the Great Lakes might play an important role in regional cloud ice formation and precipitation. Future modeling studies incorporating LSA chemical speciation and aerosol-cloud interaction are needed to shed more light on this role.





## 4  Conclusions

In this paper, we developed the first LSSF by simulating air entrainment in synthetic freshwater using a MART system. To compare freshwater and saltwater emissions, we also developed an SSSF in the same setup. Over the 0.01-10 $\mu$m particle size range, the measured total SSA number concentration was on average eight times higher than that of LSA. We show that drying the spray aerosols prior to sampling is not trivial, and leads to considerable particle losses especially in the supermicron size range. Losses in the aerosol surface area and volume size distributions, which are highly dependent on supermicron particle number concentration, are even more manifest. Therefore, it is important to take such losses into account in any experimental setup that incorporates diffusion dryers. The LSSF and SSSF developed from these experiments reveal that, at the same wind speed, LSA emissions are almost one and two orders of magnitude lower than SSA emissions for $r_{80}<0.2$ $\mu$m and $r_{80}$ ∼0.2-2 $\mu$m, respectively. Under the assumption of dry particles with a density of 1.5 gcm$^{-3}$, the emission mass flux computed from the present LSSF is at least one order of magnitude lower than that computed from the SSSFs at any wind speed.

We also implemented the developed LSSF and SSSF in the CMAQ model to examine spray aerosol emissions from the Great Lakes surface during 10 to 30 November, 2016. During an episode of very high wind speeds on 19 November 2016 at 15:00:00 UTC, LSA emissions from the Great Lakes surface reached up to $10^5$ m$^{-2}$ s$^{-1}$. The impacts of this emissions on regional aerosol number and mass concentrations were also assessed under the assumption of chemically-inert particles with a fixed density of 1.5 gcm$^{-3}$. While total aerosol number concentrations increased only by up to 1.65%, coarse-mode particle concentrations exhibited a significant 19-fold increase over northwestern Lake Superior. Looking at the mass concentration of emitted LSA particles reveals that it is dominated by coarse-mode particles and that these particles are mostly concentrated in the source region, yet they can get transported further inland up to ∼1000 km from the nearest Great Lakes source. This inland transport of coarse LSAs can have significant implications on the respiratory health of affected communities, since the enrichment of LSA particles in biological material increases with particle diameter greater than 0.5 $\mu$m (May et al., 2018b). LSA emissions also led to a significant increase in PM$_{10}$ concentrations in the region, which rose by up to 117% above the Great Lakes surface. Looking at the vertical distribution of aerosol number concentrations, our simulation shows that LSA particles reached the cloud layer, yet they only resulted in a slight (<5%) increase in total aerosol number concentrations above the lakes. However, coarse-mode particles exhibited a more significant increase of up to 144% in the layer extending up to a 1000 m above ground level, and up to a 98% increase in the cloud layer. Given the importance of supermicron particles for ice nucleation (Creamean et al., 2019; Mitts et al., 2021), this marked increase in coarse particles as a result of LSA emissions hints at possible implications on regional cloud processes.

This study highlights the errors brought about by using a SSSF to represent freshwater LSA emissions. For the case of Great Lakes emissions, for instance, using an SSSF resulted in around one order of magnitude overapproximation of LSA contribution to regional aerosol numbers and mass concentrations. Although this study laid the groundwork for future modeling studies involving LSA emissions from freshwater, it is important to note that the LSSF developed herein does not incorporate other lake conditions such as LST and biological activity. These factors are especially relevant for the Great Lakes which experience significant seasonal LST changes (Notaro et al., 2013) and episodic events of algal bloom occurrences (Bridgeman





et al., 2013). Moreover, the simulation conducted here was only based on a three-week period in November 2016, and misses LSA chemistry and LSA-cloud interaction representations. Therefore, future avenues of research include incorporating LST and biology effects into the developed LSSF, increasing the simulation period to explore seasonal emission patterns, and incorporating LSA chemistry and LSA-cloud interaction representations to better understand the effects of LSAs on regional aerosol loading and cloud processes.

*Code availability.* CMAQ source code is freely available via https://github.com/usepa/cmaq.git. Archived CMAQ versions are available from the same repository. Model input data are available from the Community Modeling and Analysis System (CMAS) Data Warehouse (https://doi.org/10.15139/S3/MHNUNE). The CMAQ simulations performed herein are reproducible using the modified CMAQ scripts available via the Virginia Tech data repository at https://doi.org/xxxxxxxxxxxxx.

*Data availability.* The experimental data used in this study are available via the Virginia Tech data repository at https://doi.org/xxxxxxxxxxxxx.

*Author contributions.* CH and HF conceptualised and designed this study. CH carried out the experiments and performed the simulations. CH analysed the data, produced the figures, and wrote the manuscript. HF provided insights on the analyses and edited the manuscript.

*Competing interests.* The authors declare that they have no conflict of interest.

*Acknowledgements.* Part of this work was performed in shared facilities at the Virginia Tech National Center for Earth and Environmental Nanotechnology Infrastructure (NanoEarth), a member of the National Nanotechnology Coordinated Infrastructure (NNCI), supported by NSF (ECCS 1542100 and ECCS 2025151). We thank Brett Gantt from the U.S. Environmental Protection Agency for assistance with the CMAQ sea spray aerosol emission scheme. We also thank Xinyue Huang from Virginia Tech for helping with setting up the CMAQ model and Nora AlAmiri from Virginia Tech for helping with water samples collection and foam images processing.



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
