# Peer review of "Experimental development of a lake spray source function and its model implementation for Great Lakes surface emissions"

_Atmospheric Chemistry and Physics, 2022_

## Author Comment (AC1)

**Reply on RC1:**

*In this article, Harb and Foroutan describe a measurement and complementary modeling study of lake spray emissions focused on the Great Lakes region. The measurement section of the article describes the development of a lake spray source function using a marine aerosol reference tank, while the modeling section uses the Community Multiscale Air Quality model to simulate the lake spray impacts on aerosol number and mass concentrations during a period of strong winds. Generally, I find the measurement section of the article a useful development in the field of lake spray emissions but find the modeling section incomplete. Please see the specific comments below.*

We thank the reviewer for taking the time to review our manuscript. Our reply to each comment is provided below.

*Major Comments:*

*1) CMAQ model: Despite being a widely-used model for research and policy, I don't believe that CMAQ is the appropriate tool to simulate the potential cloud impacts from lake spray aerosols. The lack of aerosol-cloud interactions in the version of CMAQ used in this study means that the simulations are limited to the prediction of aerosol number concentrations without any information on the associated changes of cloud condensation nuclei or ice nuclei on cloud properties. Quantifying the impacts of this aerosol source on cloud properties requires an online-coupled model.*

We agree with the reviewer that a more comprehensive evaluation of the impact of this aerosol source on cloud processes requires the use of an online-coupled model. However, the main goal of this manuscript was to report an experimentally-developed LSA emission parameterization and provide a preliminary estimate of freshwater emissions impacts on regional aerosol loading in the Great Lakes region. Therefore, we believe that implementing the LSA parameterization in an online-coupled model is beyond the scope of this work but is definitely worth considering in future studies that are revolved around the impacts of LSAs on cloud properties. In the original submission, we mentioned that future studies should consider LSA-cloud interactions (please see lines 545-548 in the revised manuscript). We have also mentioned this limitation when discussing the impacts of LSAs on the aerosol number concentrations in the cloud layer (i.e., Figs. 11 and 12) (lines 473-475 in the revised manuscript).

Revised text as it appears in the manuscript (lines 473-475):

However, the reader should keep in mind that a more comprehensive assessment of the potential impact of LSAs on cloud processes, specifically the associated changes in cloud condensation nuclei (CCN) and ice nucleating particles (INPs), requires the use of an online-coupled model (Zhang, 2008).

*2) CMAQ model configuration: I found little benefit from the SEA simulation to the study, and was surprised that it was included after the description of the lake spray source function having substantially lower emission fluxes than the sea spray source function. If CMAQ continues to used in this study (see comment above), I'd strongly suggest that the SEA simulation be replaced*

*with a simulation incorporating chemical speciation of the lake spray aerosol so that the impacts to regional atmospheric chemistry be quantified.*

One of the main takeaways from this work was that using a sea spray source function to represent freshwater emissions will lead to a considerable overestimation of LSA emissions (please see lines 538-540 in the revised manuscript). Therefore, we have considered the SEA simulation to highlight this overestimation. Moreover, we have already provided the rationale behind our choice of each emission scenario in the original submission (please see lines 243-246 in the revised manuscript).

We agree with the reviewer that including LSA chemical speciation is essential to better quantify the impact of LSA emissions on regional atmospheric chemistry. However, the main goal of the modeling section was to quantify the contribution of freshwater emissions to regional aerosol loading and compare it to anthropogenic emissions in the region. It is obviously true that the inclusion of LSA chemical speciation will result in a more realistic simulation, however, we believe that using inert LSA particles provides a good preliminary idea about these impacts. In the original submission, we have already discussed the limitations brought about by not including LSA chemistry in light of the study goals (please see lines 406-413 in the revised manuscript) and suggested including LSA chemical speciation in future studies (please see lines 544-548 in the revised manuscript).

*3) CMAQ model evaluation: When discussing the model results, it's unclear whether including this emission source improves model performance. I'd suggest that PM2.5 and PM10 observations from regulatory and IMPROVE sites in the region be compared to the model simulations. If chemical speciation of the lake spray aerosol is included (see comment above), I'd also suggest that PM2.5 speciation from the IMPROVE and CSN sites in the region be used in the evaluation.*

We thank the reviewer for this suggestion. We think that without the inclusion of the LSA chemical speciation (please see our reply to comment 2 above), such a comparison is not very useful. Therefore, we believe that such an analysis would be suitable for future modeling studies that include LSA chemical speciation (please see lines 548-549 in the conclusion section of the revised manuscript).

Revised text as it appears in the manuscript (lines 548-549):

The inclusion of LSA chemical speciation in the simulation will also allow for evaluating model results against PM observations from regulatory air-quality networks in the region.

*4) Typos:*

*Page 2, line 49: Should be "These distinct..."*

*Page 16, line 371: Should be "as shown in Fig. 4b..."*

*Figure 8 caption: Should be "concentrations"*

*Figure 9 caption: Should be "concentrations"*

*Figure 10 caption: Should be "concentrations"*

*Page 22, line 470: Should be "Great Lakes..."*

*Page 22, lines 478-479: Should be "further north or south..."*

*Figure 11 caption: Should be "from north to south..."*

*Page 25, line 514: Should be "impacts of these emissions..."*

We thank the reviewer for catching these typos. They were corrected in the revised manuscript.

---

## Author Comment (AC2)

**Reply on RC2:**

*ACP Review "Experimental development of a lake spray source function and its*

*model implementation for Great Lakes surface emissions"*

*Summary: This manuscript describes the development of a new source function focused on aerosol emissions from freshwater lakes. The work builds from measurements of size and number concentration in an aerosol reference tank to develop a function that it applies in the CMAQ model to look at regional impacts. The function is primarily applied to the Great Lakes due to their size, particularly their fetch. Overall the work is detailed, considers the prior literature, and generally provides a useful function for use when considering freshwater emissions in models. There are some minor issues to fix related to aerosol nomenclature and some minor questions to be addressed, but overall is solid, detailed work and should be publishable after revision.*

We thank the reviewer for going over the manuscript and for providing suggestions to improve the work. We provide our response to each comment below.

*Specific Points:*

1) *The authors in the abstract describe results with respect to fine aerosol (r < 0.2 um) and coarse aerosol (r ~ 1-2 um). Aerosol are usually defined in terms of diameter, not radius, so it would be helpful to readers to provide this in contact of diameter.*

   We have rephrased this sentence in the abstract to refer to aerosol sizes in terms of diameter as suggested. Please see lines 10-12 in the revised manuscript.

   Revised text as it appears in the manuscript (lines 10-12):

   Over the 0.01-10 µm aerosol diameter size range, the developed LSSF was around one order of magnitude lower than the SSSF and was around two orders of magnitude lower for aerosols with diameters between 1 and 3 µm.

2) *In the abstract as written it could be interpreted that the authors looked just 1-2 and < 0.2 um, which would imply that 0.2-1 um was not a part of the calculations. That size range has considerable amounts of LSA, so it would be helpful to clarify this point.*

   The sentence was modified to clarify that we looked at the 0.01-10 µm aerosol diameter size range. Please see lines 10-12 in the revised manuscript.

   Revised text as it appears in the manuscript (lines 10-12):

Over the 0.01-10 µm aerosol diameter size range, the developed LSSF was around one order of magnitude lower than the SSSF and was around two orders of magnitude lower for aerosols with diameters between 1 and 3 µm.

3) *Coarse aerosol is typically defined, at least by EPA, as 2.5-10 um. Calling 1-2 um coarse, could be confusing as this would fall in the fine PM (PM5) classification often used. IT would be helpful to explain and clarify this.*

To avoid this confusion, we have removed the term "coarse" when referring to the 1-4 µm particles in the abstract. Please see lines 10-12 in the revised manuscript. We have also revisited the terms referring to the aerosol modes in the remainder of the manuscript and made sure that they conform with the more traditional definition of aerosol modes, i.e., Aitken ($D_p$ = 0.01-0.1 µm), accumulation ($D_p$ = 0.1-2 µm), and coarse ($D_p$ > 2.5 µm) (Seinfeld and Pandis, 2016) (please see lines 328-330 in the revised manuscript).

Revised text as it appears in the manuscript (lines 10-12):

Over the 0.01-10 µm aerosol diameter size range, the developed LSSF was around one order of magnitude lower than the SSSF and was around two orders of magnitude lower for aerosols with diameters between 1 and 3 µm.

Revised text as it appears in the manuscript (lines 328-330):

Unlike the accumulation mode which is similar between the LSSF and the SSSF (0.72 vs 0.77 µm), the Aitken mode of the SSSF centered at 0.042 µm is greater than that of the LSSF, which is centered at 0.031 µm. This Aitken mode in the LSSF compares well with the Aitken mode (0.025-0.035 µm) measured above the Great Lakes surface (Slade et al., 2010).

4) *Line 30: A common comment regarding aerosolization of bacteria/cyanobacteria is that it is components of the biological species that are typically aerosolized, not an intact bacterium. Please modify wording to reflect that the most prominent LSA sizes there would not be intact bacteria.*

We agree with the author that intact bacteria would probably not be incorporated in submicron LSAs. The wording of this sentence was modified accordingly (please see lines 29-31 in the revised manuscript). However, we note that intact bacteria can indeed be incorporated into supermicron SSA particles (Patterson et al., 2016).

Revised text as it appears in the manuscript (lines 29-31):

Nonetheless, recent research has shown that LSA emission might pose a risk to respiratory health by being a vector for the water-to-air dispersal of biological material from freshwater bacteria (Harb et al., 2021), including cyanobacterial toxins from harmful algal blooms (HABs) (May et al., 2018b; Olson et al., 2020; Plaas and Paerl, 2021).

5) *The issue of low organic material in the simulated lake water and possibly from Claytor Lake is noted, it would be good to mention again in the conclusions as there is potential for fairly different emissions when more organics are present as is common across the Great Lakes, particularly in summertime.*

We have mentioned this point again in the conclusion section of the revised manuscript and discussed the implications on Great Lakes LSA emissions (please see lines 508-513 in the revised manuscript).

Revised text as it appears in the manuscript (lines 508-513):

There were no significant differences in LSA generation between the inorganic synthetic freshwater and Claytor Lake freshwater which might be due to low organic content in the collected Claytor Lake samples. However, it was observed that organic material might enhance LSA generation in freshwater (Olson et al., 2020). Therefore, actual LSA emissions from the Great Lakes might deviate from what is predicted in this study especially in spring and summer seasons when the lakes exhibit higher concentrations of organic material (Minor et al., 2019).

6) *A key issue in the method section is that the authors refer to "mobility diameter" when they mean "electrical mobility diameter". Mobility diameter in the aerosol world has a different definition not related to an aerosol's movement in an electric field (see Ch. 3 and 5 of Hinds Aerosol Textbook). To avoid confusion for readers this should be referred to as $d_{em}$.*

We agree with the reviewer that electrical mobility diameter is a more accurate term for the particle diameter measured by the SMPS. We have accordingly changed the term "mobility diameter" to "electrical mobility diameter" in the revised manuscript.

7) *Stitching together of SMPS and APS data would be preferable using the approach detailed in Khylstov 2004 AS&T, as opposed to removing that data. The sharp transion on Figure 2 at the SMPS/APS transition is notable and could be fixed by this approach.*

We have actually used the procedure detailed in Khlystov et al. (2004) to stitch the data (please see lines 140-142 in the revised manuscript). We think that the sharp transition in Fig. 2, which is observed mostly for wet aerosols and not for dry aerosols, is due to water accumulation at the inlet of the SMPS impactor which leads to sampling losses in its upper size range ($d_{em}$~0.5-0.7 µm). We have already discussed these losses in the original submission (please see lines 311-313 in the revised manuscript). Because of these losses and undercounting of the SMPS and APS in the overlapping size range (Stokes et al., 2013), we only removed the data in the overlapping size region when stitching (please see lines 150-151). Please note that the stitching procedure used in this study leads to a dry SSA size distribution shape that agrees well with other measurements in the MART (e.g., (Stokes et al., 2013)) and without a notable transition as seen for wet SSAs (please see lines 258-260 in the revised manuscript).

8) *The wet versus dry size distributions are interesting. Could an inset or additional panel be included to make it easier to see the wet freshwater distribution in Figure 2a, it is almost impossible to see with how big the wet saltwater is.*

We thank the reviewer for this suggestion, and we agree that it will improve the clarity of the figure. Figure 2a was modified and now includes two insets showing the LSA size distributions from freshwater samples.

9) *In a similar vein, for Figure 3 could the red size distributions (S and V) for 3c, 3d, 3e, and 3f be placed on the right axis or an inset provided? They are very difficult to distinguish from the baseline.*

Figures 3c, 3d, 3e, and 3f were also modified to include an inset.

10) *Page 13, Line 315-316: please fix the sig figs for the numbers on these two lines. Error should not be listed with more than one digit (i.e. 21722 +/- 1964 should be 22000 +/- 2000).*

The numbers were modified to include two significant digits in the average value and one significant digit in the error. Please see lines 315-318 in the revised manuscript.

Revised text as it appears in the manuscript (lines 315-318):

Indeed, the peak in surface area concentration for $D_P>1$ µm drops from 22000 ($\pm$ 2000) to 520 ($\pm$ 70) $\mu m^{-2} cm^{-3}$ in saltwater and from 250 ($\pm$ 50) to 5.4 ($\pm$ 0.8) $\mu m^{-2} cm^{-3}$ in freshwater. Similarly, the peak in volume size distribution drops from 12000 ($\pm$ 1000) to 120 ($\pm$ 30) $\mu m^{-3} cm^{-3}$ in saltwater, and from 210 ($\pm$ 20) to 1.8 ($\pm$ 0.3) $\mu m^{-3} cm^{-3}$ in freshwater.

11) *In the simulations for Figures 6-9, are larger particles even more enhanced (e.g. 3, 4, 5 um?) in the more traditional coarse definition?*

CMAQ treats aerosol size distribution as the superposition of three lognormal modes: Aitken, accumulation, and coarse (Binkowski and Roselle, 2003). Each aerosol emission source, whether anthropogenic or natural, is characterized consistently by a unique lognormal size distribution (i.e., with a specific geometric mean and geometric standard deviation for each mode). On a dry diameter basis ($D_{dry}$) (see Fig. 4b), the LSSF and SSSF are characterized by a mode at the lower end of the accumulation size range (~0.1 µm) and a second mode at the higher end of the coarse size range (~2-3 µm). Therefore, sea spray (and lake spray) aerosol emissions in CMAQ are represented as the sum of an accumulation and coarse lognormal modes. To answer the reviewer's question, in a general sense, the coarse mode does include particles with diameters greater than 3 µm and hence these particles are also enhanced in the simulation.

12) *Figure 10, why in the lake case is there so much aerosol over the Atlantic Ocean? Particularly for the $PM_5$?*

The increase in PM over the Atlantic Ocean in both the LAKE and SEA scenarios (particularly for $PM_{10}$, Figs. 10f, and 10i) is due to "an episodic event of high particle emissions from the Great Lakes surface on 20-21 November 2016, followed by atmospheric transport to the western Atlantic Ocean which is otherwise an area with low preexisting PM concentrations (Fig. 10a-c)" (lines 467-469 in the revised manuscript). Also, please note that the colormap for PM10 concentrations is different than that for PM1 and PM2.5, consequently, we have updated the legend of Figs.8 and 10 to draw the reader's attention to the difference in colormaps (please see Figs.8 and 10 legends in the revised manuscript). Therefore, it might seem from Fig. 10 that the increase in aerosols over the Atlantic Ocean is more pronounced for PM2.5, however, please note that the highest increase is for PM10 (lines 465-467). Finally, we have revisited the method that we used to calculate PM in Fig. 10. With the new method, the percent increase in PM concentration in this figure is slightly different, therefore, the associated text in the manuscript was updated (please see lines 450-471 and updated Fig. 10 in the revised manuscript).

Revised text as it appears in the manuscript (lines 450-471):

To put these mass emissions into perspective, Fig. 10 shows regional $PM_1$, $PM_{2.5}$, and $PM_{10}$ surface-layer concentrations in the BASE scenario averaged over the simulation period (a-c) and their corresponding percent increase in the LAKE (d-f) and SEA scenarios (g-i). In the BASE scenario, $PM_1$, $PM_{2.5}$, and $PM_{10}$ concentrations above the Great Lakes surface were highest over the southern parts of Lake Michigan and Lake Erie where they were about 7, 9, and 15 $\mu gm^{-3}$, respectively, and were lowest in the remote northern lakes, specifically over northwestern Lake Superior where they were about 3,4, and 6 $\mu gm^{-3}$, respectively (Fig. 10a-c). PM concentration hotspots can be clearly seen in the Chicago and Toronto metropolitan areas where $PM_1$, $PM_{2.5}$, and $PM_{10}$ average concentrations reach 15, 20, and 28 $\mu gm^{-3}$, respectively. When LSA emissions from the surface of the Great Lakes are enabled (i.e., LAKE scenario), $PM_1$ and $PM_{2.5}$ increase by up to 4% and 14% in northwestern Lake Superior, respectively, driven by low preexisting PM concentrations in that area (Fig. 10d-e). The average percent increase in $PM_{10}$ is rather more significant reaching 99 % over northwestern Lake Superior (Fig. 10f). Overall, the increase in PM concentrations in the LAKE scenario is mostly in the source region with some increase inland specifically in the vicinity of the lakes. On the other hand, SSA emissions (i.e., in the SEA scenario) result in up to 47%, 400%, and 3200% average increase in $PM_1$, $PM_{2.5}$, and $PM_{10}$ concentrations in the source region, respectively (Fig. 10g-i). Therefore, using an SSSF to represent LSA emissions resulted in one order of magnitude overestimation of Great Lakes surface emission contribution to regional PM concentrations. Interestingly, it can also be seen from Fig. 10 that the effect of surface emissions from the Great Lakes can extend far beyond the source region and into the Atlantic Ocean. For instance, the $PM_{10}$ average concentrations over the western Atlantic Ocean increased by up to 5 % and 40 % in the LAKE (Fig. 10f) and SEA (Fig. 10i) emission scenarios, respectively. These increases in faraway regions stem from an episodic event of high particle emissions from the Great Lakes surface on 20-21 November 2016, followed by

atmospheric transport to the western Atlantic Ocean which is otherwise an area with low preexisting PM concentrations (Fig. 10a-c). Therefore, emissions from the Great Lakes surface can extend further inland during episodic events of very high wind speeds—a feature that is concealed when averaging over several weeks.

**References**

Binkowski, F. S. and Roselle, S. J.: Models-3 Community Multiscale Air Quality (CMAQ) model aerosol component 1. Model description, Journal of geophysical research: Atmospheres, 108, 2003.

Khlystov, A., Stanier, C., and Pandis, S. N.: An Algorithm for Combining Electrical Mobility and Aerodynamic Size Distributions Data when Measuring Ambient Aerosol Special Issue of Aerosol Science and Technology on Findings from the Fine Particulate Matter Supersites Program, Aerosol Science and Technology, 38, 229-238, 10.1080/02786820390229543, 2004.

Patterson, J. P., Collins, D. B., Michaud, J. M., Axson, J. L., Sultana, C. M., Moser, T., Dommer, A. C., Conner, J., Grassian, V. H., and Stokes, M. D.: Sea spray aerosol structure and composition using cryogenic transmission electron microscopy, ACS Central Science, 2, 40-47, 2016.

Seinfeld, J. H. and Pandis, S. N.: Atmospheric chemistry and physics : from air pollution to climate change,  2016.

Stokes, M. D., Deane, G. B., Prather, K., Bertram, T. H., Ruppel, M. J., Ryder, O. S., Brady, J. M., and Zhao, D.: A Marine Aerosol Reference Tank system as a breaking wave analogue for the production of foam and sea-spray aerosols, Atmos. Meas. Tech., 6, 1085-1094, 10.5194/amt-6-1085-2013, 2013.